# Characterization of the bacteriophage vB_KpnP_Henu1_3 lytic for K1 *Klebsiella pneumoniae* and its therapeutic efficacy in *Galleria mellonella* larvae and mice

Yuan Zhang,[1] Lin Shi,[1] Fang Zhou,[2,3] Jiaqi Li,[2,3] Mengzhe Liu,[2,3] Shuai Guo,[2,3] Xiaoyu Shi,[2,3] Xinwei Zhang,[2] Dongliang Qiao,[2] Jiangfeng Zhang,[1] Kexiao Wang,[3] Tieshan Teng,[3] Youhua Yuan,[1] Qiming Li,[2,3,4] Shanmei Wang[1]

**ABSTRACT**  New anti-infective therapies are urgently needed for the treatment of drug-resistant bacterial infections in the context of the rapid spread of drug resistance. Phages, the natural enemies of bacteria, have irreplaceable advantages in the treatment of bacterial infections. Here, we report a novel phage, vB_KpnP_Henu1_3, that specifically lyses capsule-type K1 *Klebsiella pneumoniae*. The phage vB_KpnP_Henu1_3 shows relatively favorable thermal stability (4°C–55°C) and pH tolerance (pH = 4–11). In addition, the optimal multiplicity of infection is 0.01, with a burst size of approximately 253 ± 54 PFU/cell. Genomic analysis reveals that phage vB_KpnP_Henu1_3 contains double-stranded DNA (total length of 49,808 bp) with a G + C content of 50.76%, and the genome comprises 75 open reading frames with no virulence- or antibiotic resistance-related genes. Transmission electron microscopy observations revealed that phage vB_KpnP_Henu1_3 possessed an icosahedral head and siphovirus morphology. Based on the genome sequence, phage vB_KpnP_Henu1_3 could be assigned to a new species in the genus *Webervirus* of the family *Drexlerviridae*. Furthermore, phage vB_KpnP_Henu1_3 rapidly inhibits the growth of *K. pneumoniae* within 4 h, prevents biofilm formation, and disrupts mature biofilms *in vitro*. In infection animal models, phage vB_KpnP_Henu1_3 significantly increases the survival rate of *K. pneumoniae*-infected *Galleria mellonella* larvae and mice while reducing the bacterial loads. These findings demonstrate that phage vB_KpnP_Henu1_3 has promising potential as a safe alternative for controlling and treating multidrug-resistant K1-type *K. pneumoniae* infections.

**IMPORTANCE** The widespread use of antibiotics has led to increasing antibiotic resistance, which is a growing global health concern. Therefore, the development of novel antimicrobial therapy that can cure drug-resistant bacteria-induced infections is imperative. Phages are of increasing interest as natural enemies of bacteria with clear advantages in antibacterial.

**KEYWORDS**  bacteriophage, phage therapy, antibiotic resistance, *Klebsiella pneumoniae*, vB_KpnP_Henu1_3

*K*lebsiella pneumoniae has emerged as a clinically significant pathogen, accounting for a substantial proportion of both healthcare-associated infections and community-acquired invasive diseases. As a leading etiological agent, it demonstrates particular clinical relevance in hospital settings where it frequently causes pneumonia, bloodstream infections, and urinary tract infections among immunocompromised patients. *K. pneumoniae* has evolved from having typical characteristics to being a highly virulent *K. pneumoniae* (hvKp) and is widely prevalent worldwide (1, 2). With the widespread use of broad-spectrum antimicrobials such as β-lactams and

**Peer Reviewer** Ramzi Atiah Alahmadi, King Abdulaziz University, Jeddah, Makkah al mukarramah, Saudi Arabia

Address correspondence to Qiming Li, liqiming82@126.com, or Shanmei Wang, wsm1997@zzu.edu.cn.

The authors declare no conflict of interest.

See the funding table on p. 18.

aminoglycosides, *K. pneumoniae* is prone to produce extended-spectrum β-lactamases (ESBLs) and cephalosporinases (AmpC), as well as aminoglycoside-modifying enzymes, and exhibits severe multidrug resistance to commonly used drugs, including third-generation cephalosporins and aminoglycosides (3–6). The prevalence of multidrug resistance genes has been accompanied by an increasing number of multidrug-resistant, highly virulent *K. pneumoniae* strains (7). The most common serotypes of hvKp include K1, K2, K20, K54, and K57, with K1 and K2 being the most virulent and accounting for 70% of hvKp isolates (8). The prevalence of the mobile colistin resistance 1 plasmid has also allowed *K. pneumoniae* to break through the last resort of antibiotics against negative bacterial infections, colistin (9). Cases of hospital-acquired infections caused by *K. pneumoniae* have been increasing annually. Additionally, the increasing number of multidrug-resistant strains frequently leads to the failure of clinical antimicrobial therapy and the prolongation of disease. There is an imperative clinical need for innovative therapeutic approaches for the management of *K. pneumoniae* infections (10).

Phages, which serve as the natural predators of bacteria, have irreplaceable advantages over antimicrobial agents and are employed in the treatment of bacterial infections (11). Currently, with the widespread prevalence of drug-resistant bacterial infections, the clinical use of phage therapy has witnessed an increase (12). The pathways by which antibiotics and phages cause bacterial death are completely different, so bacteria that are resistant to antibiotics do not affect bacterial susceptibility to phages (13). There have been many reports of effective phage therapy for *K. pneumoniae* infection. For example, Gina et al. administered intravenous phage therapy to a 62-year-old patient with persistent infection of the right prosthetic knee caused by the *K. pneumoniae* complex, and this therapy resulted in the resolution of local symptoms and signs of infection (14). Mario et al. used a lytic bacteriophage preparation to treat a patient with *K. pneumoniae* infection that produced multidrug-resistant carbapenemase (KPC-3) and resulted in the eradication of the microorganism without adverse effects (15). Bacteria have developed an astonishing array of strategies to cope with phage threats at each step of the infection process. When bacteria are infected, they can defend against phages by preventing phage adsorption, preventing phage DNA entry, cutting phage nucleic acids, and activating abortive infection systems (16). Of course, bacteria that have been in contact with one phage for a long time promise to foster adaptive evolution and thus be resistant to the phage. However, this adaptive evolution will come at a cost in terms of renewed sensitivity to antibiotics. Therefore, the combined use of antibiotics and phages is also an effective way to address drug-resistant bacterial infections and combat the emergence of bacterial drug resistance. Saskia et al. reported that a 58-year-old kidney transplant patient infected with ESBL-positive *K. pneumoniae* had no response to repetitive treatment with meropenem but was successful with a combination of meropenem and phages (17).

Host specificity and the generation of phage resistance limit the clinical application of phage therapy. The widespread presence of phages in the environment provides many raw materials for phage cocktails. Therefore, the establishment of phage libraries by the isolation and identification of different species of phages and the study of their characteristics are important for the future clinical application of phages. Phages infecting *K. pneumoniae* have been isolated from a variety of environments, including sewage, sludge, and infected samples from animals and humans. However, the number of *K. pneumoniae* phages that have been isolated and characterized is insufficient for future clinical applications. Therefore, it is imperative to isolate a large number of phages as stockpiles for the treatment of future clinically resistant bacterial infections. Hospitals, as aggregation and transmission transit sites for drug-resistant bacteria, may harbor large numbers of phages capable of infecting drug-resistant *K. pneumoniae*.

In this study, we isolated a *K. pneumoniae* phage, named vB_KpnP_Henu1_3, from sewage samples from Kaifeng pulmonary hospital. Biological characteristics, genomic features, and antimicrobial efficacy *in vitro* and *in vivo* were evaluated to

determine the application value of vB_KpnP_Henu1_3. Our results indicated that phage vB_KpnP_Henu1_3 has excellent antimicrobial capacity both *in vitro* and *in vivo* and has great potential for clinical application.

## MATERIALS AND METHODS

### Phage isolation and purification

The bacteriophage was isolated following a previously described method with modifications (18). Briefly, untreated sewage collected from Kaifeng Pulmonary Hospital was subjected to initial centrifugation and filtration through a 0.22 μm membrane. The resulting supernatant was combined with *K. pneumoniae* strain Kp1049 (isolated in December 2023, accession number CCTCC PB 2025034) and 0.7% soft agar, which was then overlaid onto LB solid medium. After 24 h of incubation, a single, well-isolated phage plaque was aseptically picked and eluted in PBS buffer. The eluate was subsequently spotted onto a fresh lawn of *K. pneumoniae* Kp1049 for successive rounds of plaque purification. This isolation process was repeated five times to ensure homogeneity and stability of the phage population. Through this stringent purification protocol, a novel lytic bacteriophage specific to *K. pneumoniae* was successfully isolated and designated as *Klebsiella* phage vB_KpnP_Henu1_3.

### Transmission electron microscopy analysis

The morphology of purified phage vB_KpnP_Henu1_3 was examined by negative-stain transmission electron microscopy (TEM). Briefly, 20 μL of high-titer phage lysate was adsorbed onto a carbon-coated copper grid (300 mesh) for 10 min at room temperature. Excess liquid was carefully blotted away, and the grid was negatively stained with 2% (wt/vol) uranyl acetate for 90 s. After air-drying, samples were imaged using a Hitachi TEM system operating at an accelerating voltage of 80 kV.

### Host range of phage vB_KpnP_Henu1_3

Twenty-seven clinical isolates of *K. pneumoniae* and other gram-negative bacteria were collected from various hospital ward environments. The detailed information of all clinical isolates of *K. pneumoniae* is listed in Table S1. All isolates were cultured to the mid-exponential phase prior to analysis. For phage susceptibility testing, bacterial cultures were mixed with 0.7% molten soft agar and overlaid onto LB agar plates to create double-layer agar assays. Ten-fold serial dilutions of phage suspension were spotted onto the prepared bacterial lawns, followed by incubation at 37°C for 12 h. Plaque formation was then assessed to determine phage susceptibility. Efficiency of plating (EOP) is calculated in percent as the PFU/mL of the phages on the test strain divided by the PFU/mL obtained on strain Kp1049 multiplied by 100.

### Determination of optimal multiplicity of infection

Bacterial cells were cultured to the logarithmic phase ($OD_{600} \approx 0.5$), harvested, and resuspended in fresh LB medium at an initial concentration of $1 \times 10^8$ CFU/mL. Phage vB_KpnP_Henu1_3 was then added at varying multiplicity of infections (MOIs; 100, 10, 1, 0.1, 0.01, 0.001, and 0.0001), followed by a 10-min adsorption period at 37°C with gentle agitation. Unadsorbed phages were removed by centrifugation ($9,000 \times g$, 5 min), and the infected bacterial pellets were resuspended in 5 mL of fresh LB medium. The cultures were further incubated with shaking (200 rpm) at 37°C for 2 h to allow phage replication. After incubation, the samples were centrifuged ($12,000 \times g$, 10 min), and the supernatants were filtered (0.22 μm pore size) to remove residual bacterial cells. Finally, phage titers were determined via the standard double-layer plaque assay.

## Phage adsorption assay

The adsorption kinetics were analyzed using a modified version of established protocols (18). Briefly, *K. pneumoniae* Kp1049 was grown to the mid-logarithmic phase ($OD_{600} \approx 0.5$) prior to infection. Phage particles were introduced at an MOI of 0.01 and incubated with shaking at 37°C. Aliquots (200 µL) were collected at defined intervals (0, 2, 4, 6, 8, 10, and so on), immediately centrifuged ($12,000 \times g$, 2 min), and the supernatant was serially diluted in PBS. Residual unadsorbed phage titers were determined in triplicate using the double-layer agar method. The adsorption rate was calculated as follows: Unadsorption rate (%) = [(free phage titer)/(initial phage titer)] × 100.

## One-step growth curve analysis

The one-step growth curve was determined to measure the incubation period and the burst size of the phage as previously reported with some modifications (19, 20). Briefly, exponential-phase *K. pneumoniae* Kp1049 ($1 \times 10^8$ CFU/mL) was infected at an MOI of 0.1 and incubated at 37°C for 10 min to allow phage adsorption. Following a brief centrifugation ($4,000 \times g$, 5 min, room temperature) to remove unadsorbed phages, the pellet was resuspended in pre-warmed LB broth at a 1:1,000 dilution. The infected culture was then incubated at 37°C with shaking (180 rpm). Aliquots were collected at 10-min intervals over a 120-min period, immediately centrifuged ($12,000 \times g$, 1 min) to separate phage particles from bacterial debris. Phage titers at each time point were subsequently quantified using the standard double-layer agar plaque assay. The experiment was repeated on at least three separate occasions. Phage burst size was calculated as described previously (20).

## Temperature and pH stability determination

To measure the stability of the phage, the titer of phage vB_KpnP_Henu1_3 at different pH values and temperatures was determined. For temperature stability, 100 µL of purified high-titer phages was incubated in a water bath at 4°C, 25°C, 37°C, 45°C, or 55°C for 12 h. Then, the phage titer was determined via the double-layer plate method. For pH stability, equal amounts of phage vB_KpnP_Henu1_3 were incubated at pH values of 2, 3, 4, 5, 6, 7, 8, 9, 10, 11, and 12. The pH stability of phages was assessed by incubating them in PBS solutions adjusted to specific pH levels using hydrochloric acid (HCl). Briefly, 1% (vol/vol) of phage suspension was added to each pH-adjusted PBS solution and incubated at 4°C for 12 h. The remaining phage titers were then determined using the double-layer agar method. All the experiments were repeated three times.

## Phage DNA isolation and sequencing

The genomic DNA of phage vB_KpnP_Henu1_3 was extracted using a standard protocol as previously described (21, 22). Briefly, the purified phage vB_KpnP_Henu1_3 was incubated with DNase I (5 µg/mL) and RNaseA (1 µg/mL) to remove the host DNA. Subsequently, EDTA (pH = 8.0, 20 mM), proteinase K (50 µg/mL), and SDS (0.5% vol/vol) were added to digest the nucleocapsid of the phage and release the phage DNA. The DNA was then purified using Tris-phenol and chloroform, followed by ethanol precipitation. Finally, the pellet was dissolved in distilled water used for DNA sequencing. A paired-end 150 bp DNA library was subsequently constructed via the TruSeqTM DNA Sample Prep Kit according to the manufacturer's instructions, and the phage vB_KpnP_Henu1_3 genome was sequenced on the Illumina NovaSeq platform. The raw sequencing data were filtered for low-quality reads (≤50 bp) and adapters via Trimmomatic 0.36 with default parameters (23). The quality threshold applied for retaining reads (including those longer than 50 bp) requires an average Q value of ≥20 across 5-bp sliding windows. This criterion ensures that the filtered reads not only satisfy the length requirement but also correspond to a sequencing error rate of ≤1%. Finally, the clean reads were assembled to form a 49,808 bp circular contig via A5-MiSeq and SPAdes software (24, 25).

## Bioinformatics analysis

Open reading frames (ORFs) were predicted via Softberry (http://www.softberry.com/). The ORFs were annotated via the protein basic local alignment search tool (BLASTp) of the National Center for Biotechnology Information (NCBI) server with standard databases. The putative transfer RNA genes in the phage vB_KpnP_Henu1_3 genome were determined by tRNAscan-SE (26). The possible virulence and pathogen genes carried by the phage genome were predicted with VirulenceFinder. Antimicrobial resistance genes and lifestyle traits were predicted using PhageScope (27). The classification of phage vB_KpnP_Henu1_3 was performed using the TaxMyPhage tool in accordance with ICTV guidelines (28). Graphical maps of the annotated phage vB_KpnP_Henu1_3 genome were prepared using Proksee (29).

## Comparative genome analysis

For multiple comparisons, phage genomic sequences exhibiting similarity to vB_KpnP_Henu1_3 were identified and subjected to sequence similarity analysis using NCBI BLASTn (https://blast.ncbi.nlm.nih.gov/). Comparative genomes were mapped by the Easyfig 2.2.3 tool program.

## Bacteriolytic activity *in vitro*

One milliliter of *K. pneumoniae* Kp1049 in the exponential growth phase was transferred into 50 mL of LB medium. Phage vB_KpnP_Henu1_3 was added with MOIs of 1000, 100, 10, 1, 0.1, and 0.01 to infect *K. pneumoniae* Kp1049. The uninfected culture served as a positive control. The samples were collected every 0.5 h, and the $OD_{600}$ was measured via an ultraviolet spectrophotometer. To evaluate the inhibitory effects of phage vB_KpnP_Henu1_3 on biofilm formation, overnight cultures of *K. pneumoniae* Kp1049 were adjusted to $10^7$ CFU/mL in fresh LB medium. Bacterial suspensions were mixed with phage vB_KpnP_Henu1_3 at MOIs of 100, 10, 1, 0.1, and 0.01. Following a 10-min adsorption at 37°C, 200 µL aliquots were transferred to a sterile 96-well microplate ($n$ = 3). 200 µL bacterial suspension without phage as the positive control. Plates were incubated statically for 12 h at 37°C. Biofilm formation was quantified using crystal violet staining (G1059, Solarbio) as previously described (30). To assess phage-mediated disruption of established biofilms, mature biofilms were formed by incubating 200 µL aliquots of *K. pneumoniae* Kp1049 ($10^7$ CFU/mL) in 96-well plates for 24 h at 37°C. Supernatants were carefully removed, then wells were gently washed twice with PBS to remove non-adherent cells. Phage suspensions (200 µL) at the same MOIs used in inhibition assays were added to respective wells ($n$ = 3). Biofilms without phage treatment served as the positive control. Plates were re-incubated for 12 h at 37°C, and biofilm biomass was quantified using the same crystal violet staining protocol. All experiments were performed with three independent biological replicates.

## Evaluation of antibacterial ability in *G. mellonella* larvae

The *G. mellonella* larvae were chosen as a model to evaluate the antibacterial effect of vB_KpnP_Henu1_3. First, active *G. mellonella* larvae of approximately 0.3–0.5 g were selected and injected with 5 µL of *K. pneumoniae* Kp1049 at different concentrations into the last remaining leg to determine the appropriate infection dose. Ten larvae were delivered to each group, and three parallel experiments were conducted. *K. pneumoniae* Kp1049 was washed with saline and then diluted to $1 \times 10^9$ CFU/mL. After 1 h of bacterial infection, 5 µL of the phage vB_KpnP_Henu1_3 suspension was injected with the MOIs of 100, 10, 1, 0.1, and 0.01, and then incubated at 37°C in the dark. The *G. mellonella* larvae infected with *K. pneumoniae* Kp1049 and treated with PBS served as the negative control group. The survival rate of each group was recorded every 24 h for 7 days. For bacterial loads, 5 µL of a $1 \times 10^8$ CFU/mL bacterial suspension was injected to construct the *G. mellonella* larval infection model ($n$ = 6), and 5 µL of the phage vB_KpnP_Henu1_3 suspension was injected with MOIs of 100, 10, and 1 after 1 h of bacterial infection.

The bacteria were incubated at 37°C in the dark for 24 h, and the bacterial loads were calculated as CFUs.

## Evaluation of antibacterial ability in a mouse model

An experimental pneumonia mouse model was established by injecting *K. pneumoniae* Kp1049 intraperitoneally as previously reported (31). In brief, the bacteria were cultured in LB media to obtain a mixture with an $OD_{600}$ of 0.5. The bacterial cultures were subsequently centrifuged (8,500 × *g*, 5 min) and washed twice with saline. Finally, the bacteria were resuspended to $1 \times 10^{11}$ CFU/mL to determine the survival rate of the mice, and $1 \times 10^{10}$ CFU/mL bacteria were used to quantify the bacterial loads. One hundred microliters of bacterial suspension was injected into the peritoneal cavity of mice to establish a bacteremia infection model. The phage solutions were prepared with different MOIs and were injected intraperitoneally 1 h after bacterial infection, with saline injection serving as the negative control. For mouse survival, 30 female C57BL/6 N mice were intraperitoneally injected with bacteria for infection and divided into 6 groups (*n* = 5). The survival rates of the mice infected with bacteria were monitored and recorded for 7 days. For bacterial loads, 12 female C57BL/6 N mice were intraperitoneally injected with bacteria for infection and divided into 3 groups (*n* = 6). The organs of the mice (heart, liver, spleen, lung, and kidney) were collected separately after 48 h of treatment, and the bacterial loads were analyzed via colony formation experiments (CFU/g) conducted on 1.5% LB agar plates to demonstrate the therapeutic effect of phage vB_KpnP_Henu1_3 on pneumonia model mice.

## Statistical analysis

All experiments were performed with at least three independent biological replicates. Data are presented as the mean ± standard deviation (SD) unless otherwise indicated. Statistical significance was defined as *$P$ < 0.05, **$P$ < 0.01, and ***$P$ < 0.001 for all analyses. Data processing and visualization were conducted using GraphPad Prism 9.0 (GraphPad Software, Inc., Boston, MA, USA).

## RESULTS

### Isolation and morphology of phage vB_KpnP_Henu1_3

The phage was isolated from hospital sewage using the clinical *K. pneumoniae* Kp1049 as a host. The filtered effluent was applied to a lawn containing the *K. pneumoniae* Kp1049. After five rounds of successive picking of individual phage plaques, homogeneous, clear plaques with halos were observed following 12 h of incubation at 37°C (Fig. 1A). TEM analysis revealed that vB_KpnP_Henu1_3 has an icosahedral head with an average diameter of 69.13 ± 0.67 nm and a noncontractile long tail averaging 179.86 ± 2.01 nm in length (Fig. 1B).

### Host range of phage vB_KpnP_Henu1_3

Phages are strictly host-selective, so the host range of phages determines their clinical applications. *K. pneumoniae* Kp1049, the host bacterium of phage vB_KpnP_Henu1_3, is a K1-type *K. pneumoniae* strain. To test the host range of phage vB_KpnP_Henu1_3, 30 clinical isolates of *K. pneumoniae* and other gram-negative bacteria were employed for the detection of phage vB_KpnP_Henu1_3 infectivity. The K-types of *K. pneumoniae* were identified via *wzi* gene sequencing (32). The results demonstrate that phage vB_KpnP_Henu1_3 infected all the tested K1-type *K. pneumoniae* strains but failed to infect other capsular types of *K. pneumoniae, E. coli, A. baumannii,* or *P. aeruginosa* (Table 1). Furthermore, the phage vB_KpnP_Henu1_3 exhibited an EOP exceeding 70% against all tested clinical isolates of K1-type *K. pneumoniae* (Table 1). These results indicated that vB_KpnP_Henu1_3 is a *Klebsiella* phage with strict host restriction.

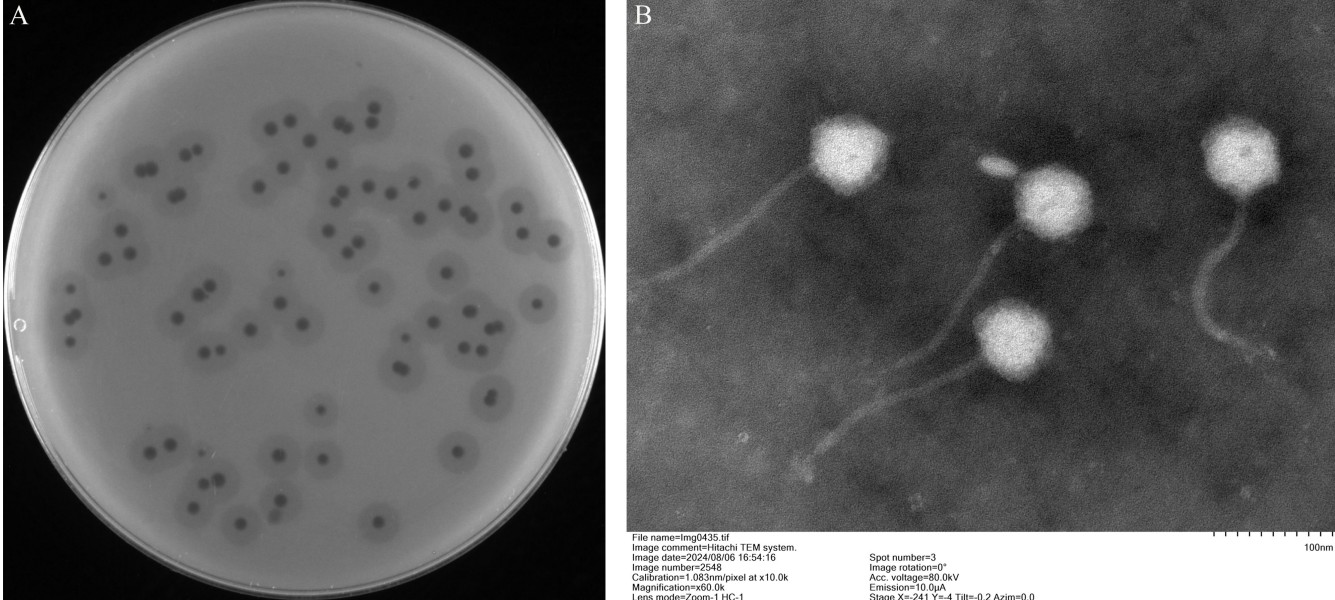

**FIG 1** Morphological observation of phage vB_KpnP_Henu1_3. (A) Phage plaques formed on the lawn of *K. pneumoniae* Kp1049 at 37°C for 12 h. (B) TEM of phage vB_KpnP_Henu1_3.

## Biological characterization of phage vB_KpnP_Henu1_3

To determine the best MOI, phage vB_KpnP_Henu1_3 was mixed with the host *K. pneumoniae* Kp1049 at different MOIs. The titer was significantly greater than that of the other groups when *K. pneumoniae* Kp1049 was infected with vB_KpnP_Henu1_3 at an MOI of 0.01, indicating that 0.01 was the optimal MOI for phage vB_KpnP_Henu1_3 (Fig. 2A). Adsorption curve analysis revealed that more than 90% of phage vB_KpnP_Henu1_3 completely adsorbed to the host *K. pneumoniae* Kp1049 after 15 min of incubation (Fig. 2B). The one-step growth curve of phage vB_KpnP_Henu1_3 was plotted on the basis of the phage titer at different incubation times. The results showed that the latent period of phage vB_KpnP_Henu1_3 was 20 min, and the phage titer gradually increased and then plateaued at 50 min. The burst size of phage vB_KpnP_Henu1_3 is approximately 253 ± 54 PFU per infected cell (Fig. 2C).

## Tolerance of phage vB_KpnP_Henu1_3 to temperature and pH

The infection stability of phages in different environments is critical for phage applications. We tested phage vB_KpnP_Henu1_3 infection in the host *K. pneumoniae* Kp1049 at different temperatures and pH values. The phage titer of vB_KpnP_Henu1_3 did not significantly change when the mixture was incubated at 4°C–55°C for 12 h (Fig. 3A). These data indicated that phage vB_KpnP_Henu1_3 could tolerate a wide range of temperatures and was suitable for use at temperatures less than 55°C. The phage vB_KpnP_Henu1_3 shows profound stability at pH values ranging from 3–11 for 12 h, while the phage titer significantly decreased at pH values below 3 or above 12 (Fig. 3B).

## The characterization of phage vB_KpnP_Henu1_3 genome

The genome of phage vB_KpnP_Henu1_3 is a double-stranded DNA with a total length of 49,808 bp and a G + C content of 50.76%, which can be digested by *Bam*HI and *Eco*RI, but not *Hin*dIII or *Xho*I (Fig. 4A). The phage vB_KpnP_Henu1_3 genome contains 75 ORF (GenBank accession number: PQ133004.1), and no tRNA genes, known drug resistance genes, or virulence factor genes were identified (Fig. 4B). Only 18 of the 75 ORFs encoded by the vB_KpnP_Henu1_3 genome were annotated as functional proteins by BLASTp, and the rest of the ORFs were annotated as hypothetical proteins (Table 2).

**TABLE 1** Host range analysis of phage vB_KpnP_Henu1_3 against 30 strains[a]

| Species | Strains | Capsular type | Susceptibility | Origin | EOP[b] |
|---|---|---|---|---|---|
| K. pneumoniae | Kp1049 | K1 | ++++ | The First Affiliated Hospital of Henan University | 100 |
| | Kp0311 | K1 | ++++ | The First Affiliated Hospital of Henan University | 92 |
| | Kp0822 | K1 | ++++ | The First Affiliated Hospital of Henan University | 89 |
| | Kp407 | K1 | ++++ | Henan Provincial People's Hospital | 76 |
| | Kp0918 | K1 | ++++ | The First Affiliated Hospital of Henan University | 83 |
| | Kp408 | K2 | – | Henan Provincial People's Hospital | 0 |
| | Kp1203 | K2 | – | The First Affiliated Hospital of Henan University | 0 |
| | Kp1616 | K2 | – | The First Affiliated Hospital of Henan University | 0 |
| | Kp0524 | K2 | – | The First Affiliated Hospital of Henan University | 0 |
| | Kp1904-2431 | K19 | – | The First Affiliated Hospital of Henan University | 0 |
| | Kp302 | K19 | – | Henan Provincial People's Hospital | 0 |
| | Kp309 | K63 | – | Henan Provincial People's Hospital | 0 |
| | Kp403 | K62 | – | Henan Provincial People's Hospital | 0 |
| | Kp2640 | K28 | – | The First Affiliated Hospital of Henan University | 0 |
| | Kp126N | K28 | – | The First Affiliated Hospital of Henan University | 0 |
| | Kp306N | K14,K64 | – | The First Affiliated Hospital of Henan University | 0 |
| | Kp347N | K14,K64 | – | The First Affiliated Hospital of Henan University | 0 |
| | Kp2001-0185 | K14,K64 | – | Henan Provincial People's Hospital | 0 |
| | Kp2001-0219 | K14,K64 | – | Henan Provincial People's Hospital | 0 |
| | Kp2011-3676 | K14,K64 | – | Henan Provincial People's Hospital | 0 |
| | Kp1901-0124 | K14,K64 | – | Henan Provincial People's Hospital | 0 |
| | Kp120804 | K62 | – | Henan Provincial People's Hospital | 0 |
| | Kp120819 | K63 | – | Henan Provincial People's Hospital | 0 |
| | Kp57N | K63 | – | The First Affiliated Hospital of Henan University | 0 |
| | Kp0953 | K19 | – | The First Affiliated Hospital of Henan University | 0 |
| | Kp0706 | K14 | – | The First Affiliated Hospital of Henan University | 0 |
| | Kp2828 | K16 | – | The First Affiliated Hospital of Henan University | 0 |
| E. coli | Ec1033 | | – | The First Affiliated Hospital of Henan University | 0 |
| A. baumannii | Ab2055 | | – | The First Affiliated Hospital of Henan University | 0 |
| P. aeruginosa | Pa3046 | | – | The First Affiliated Hospital of Henan University | 0 |

[a]++++, plaques at $10^6$; +++, plaques at $10^5$; ++, plaques at $10^4$; –, no plaques.
[b]EOP is calculated in percent as the PFU/mL of the phages on the test strain divided by the PFU/mL obtained on strain Kp1049 multiplied by 100.

From the outside to the inside of phage vB_KpnP_Henu1_3, the genomic map is listed in the order of ORF feature, GC content, and GC skew (Fig. 4B). The BLAST analysis at the whole-genome level revealed that phage vB_KpnP_Henu1_3 is homologous to eight previously described *Klebsiella* phages (Table 3), and multiple sequence comparisons are shown in Fig. 5A. Phage vB_KpnP_Henu1_3 has the highest similarity with phage RCIP0025 at 92.9% (Fig. 5B). Based on ICTV guidelines, phage vB_KpnP_Henu1_3 was identified as a new species, along with RCIP0025, RCIP0059, and KOX1, in the genus *Webervirus* of the family *Drexlerviridae*.

## Evaluation of phage vB_KpnP_Henu1_3 against *K. pneumoniae* Kp1049 *in vitro*

The strong lytic activity of phages is the basis for phage therapy used in the clinic. In this study, the lytic activity of phage vB_KpnP_Henu1_3 was measured by mixing it with *K. pneumoniae* Kp1049 in the pre-logarithmic growth phase at different MOIs and incubating it for 6 h. The growth curves revealed that treatment with phage vB_KpnP_Henu1_3 at MOIs ranging from 0.01 to 1,000 completely inhibited *K. pneumoniae* Kp1049 within 4 h (Fig. 6A). The $OD_{600}$ of each culture group gradually increased over time, suggesting that *K. pneumoniae* Kp1049 may have developed phage resistance through mutations.

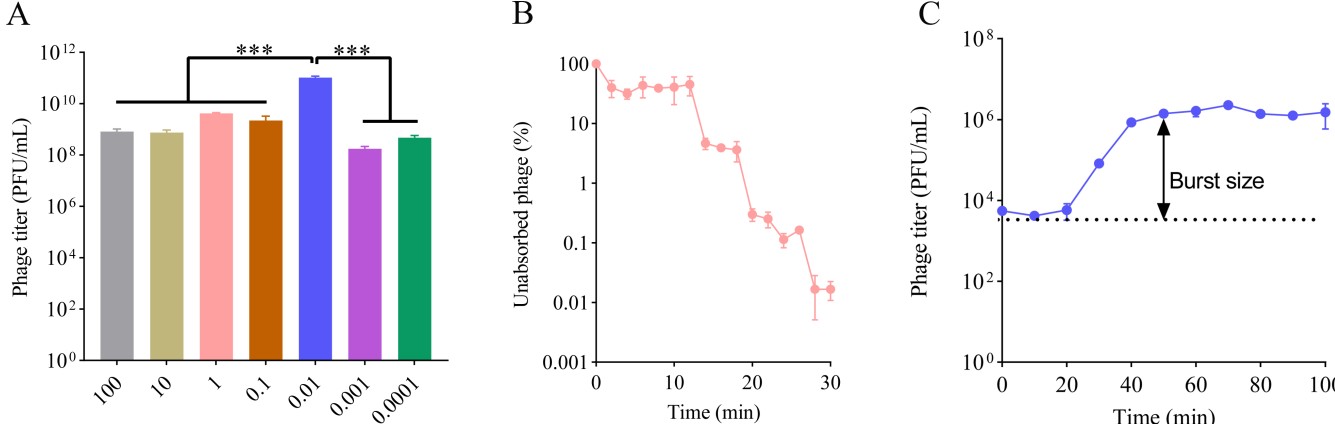

**FIG 2** Biological characterization of phage vB_KpnP_Henu1_3. (A) Determination of the optimal MOI. Bacteria were infected with phage at various MOIs, adsorbed (37°C, 10 min), centrifuged to remove unadsorbed phages, incubated for replication (37°C, 2 h), and the progeny phage titer was determined by plaque assay. (B) Adsorption kinetics of phage vB_KpnP_Henu1_3 to *K. pneumoniae* Kp1049 at an MOI of 0.01, showing the percentage of unadsorbed phages over time as determined by quantifying free phage titers in the supernatant. (C) One-step growth curve analysis of phage vB_KpnP_Henu1_3, revealing its latent period and burst size, as determined by quantifying progeny phage titers at intervals following synchronous infection of *K. pneumoniae* Kp1049. Data represent means ± SD (*n*=3), statistical analyses were conducted using Student's *t*-test for comparisons between groups. \*\*\**P* < 0.001, \*\**P* < 0.01, \**P* < 0.05.

Biofilms are one of the means by which bacteria defend themselves against harsh external environments and are highly resistant to antimicrobial agents. Biofilm formation can exacerbate infections in patients, resulting in harsh treatment and causing great concern in the healthcare environment. Biofilm formation by *K. pneumoniae* can facilitate colonization in multiple anatomical sites, including the gastrointestinal, respiratory, and urogenital tracts, and may subsequently promote the development of invasive infections (33). Several studies have reported that phage or phage-derived proteins are effective at removing biofilms (34, 35). Therefore, the ability of phages to inhibit biofilm formation or remove mature biofilms is among the important evaluation indices of phages used for therapy. In this study, we evaluated the inhibition and removal of biofilms by phage vB_KpnP_Henu1_3. Coincubation of *K. pneumoniae* Kp1049 with phage vB_KpnP_Henu1_3 at varying MOIs (0.0001–100) significantly reduced biofilm biomass (Fig. 6B). Mature biofilms were also significantly disrupted when the mature biofilms were incubated with phage vB_KpnP_Henu1_3 at MOIs ranging from 0.0001 to 100 (Fig. 6C). These results indicated that phage vB_KpnP_Henu1_3 could significantly inhibit biofilm formation and disrupt mature biofilms of *K. pneumoniae* Kp1049.

## Evaluation of phage vB_KpnP_Henu1_3 against *K. pneumoniae* Kp1049 *in vivo*

The larvae of *G. mellonella*, an animal model prone to infection by a wide range of pathogenic microorganisms, have become an attractive model for the preliminary evaluation of virulence and efficacy (36). Exploring the antimicrobial effects of phages in *G. mellonella* larvae infection models has also become one of the means of initially evaluating the therapeutic efficacy of phages *in vivo* (37, 38). In this study, we evaluated the effect of phage vB_KpnP_Henu1_3 *in vivo* on *G. mellonella* larvae infected with *K. pneumoniae* Kp1049. Each *G. mellonella* larva was infected with *K. pneumoniae* Kp1049 at a concentration of $1 \times 10^9$ CFU/mL and treated with phage vB_KpnP_Henu1_3 at MOIs of 0.01, 0.1, 1, 10, and 100 one hour later, with the PBS-treated group served as a negative control (Fig. 7A). The *G. mellonella* larvae treated with PBS died within 2 days, and the survival rates of *G. mellonella* larvae treated with phage vB_KpnP_Henu1_3 at MOIs of 0.01, 0.1, 1, 10, and 100 improved by 20%, 30%, 40%, 60%, and 60%, respectively (Fig. 7B). In addition, the loads of bacteria in *G. mellonella* larvae were significantly diminished in the phage vB_KpnP_Henu1_3 treatment group (Fig. 7C). The above results demonstrated

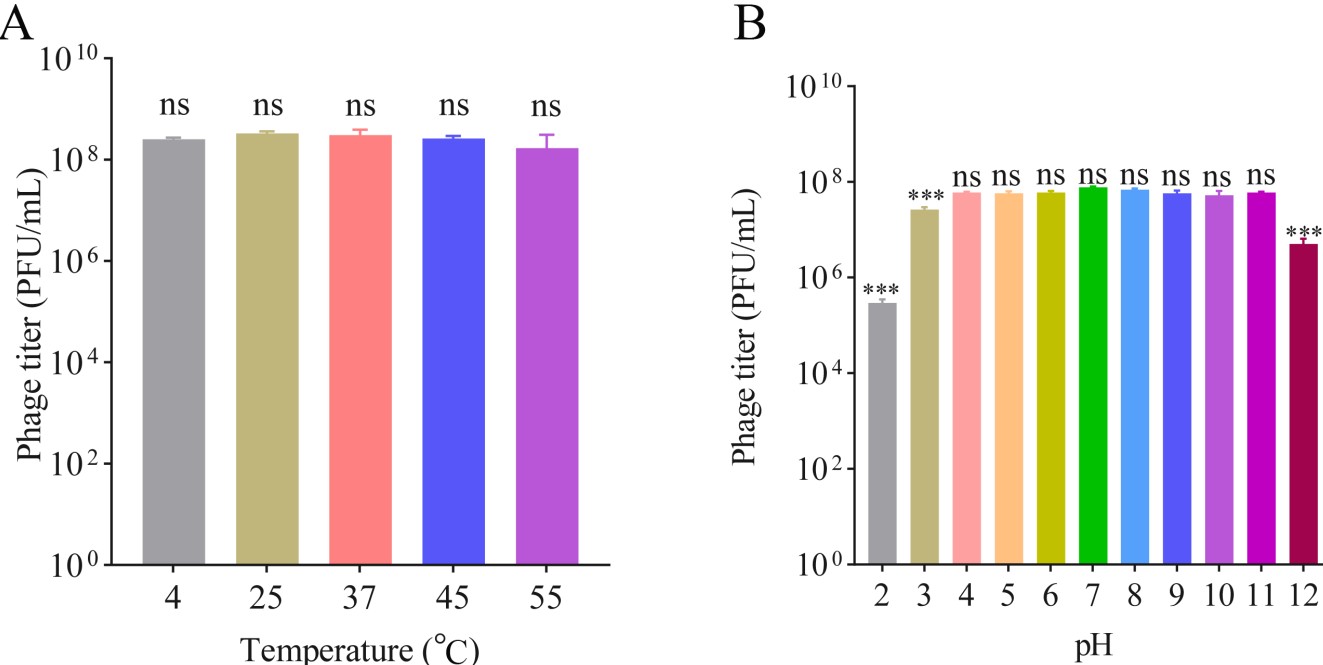

**FIG 3** Sensitivity of phage vB_KpnP_Henu1_3 to temperature and pH. (A) Thermal stability of phage vB_KpnP_Henu1_3. The purified phage vB_KpnP_Henu1_3 suspensions were incubated at 4°C, 25°C, 37°C, 45°C, and 55°C for 12 h, after which the phage titer was detected. (B) pH stability of phage vB_KpnP_Henu1_3. vB_KpnP_Henu1_3 suspensions were incubated at different pH values ranging from 2 to 12 for 12 h, after which the phage titer was detected. Data are shown as the means ± SD ($n$ = 3). Statistical differences among groups were analyzed using the Kruskal-Wallis test (a non-parametric alternative to ANOVA), followed by Dunn's post hoc test for pairwise comparisons. ***$P < 0.001$, ns, not significant.

that phage vB_KpnP_Henu1_3 significantly reduced the bacterial load of *K. pneumoniae* Kp1049 in infected *G. mellonella* larvae within 24 h.

Furthermore, the therapeutic effect of phage vB_KpnP_Henu1_3 was evaluated via the peritoneal infection of mice with *K. pneumoniae* Kp1049. The mouse peritonitis-sepsis infection model was constructed by injection of 100 µL of *K. pneumoniae* Kp1049 (1 ×

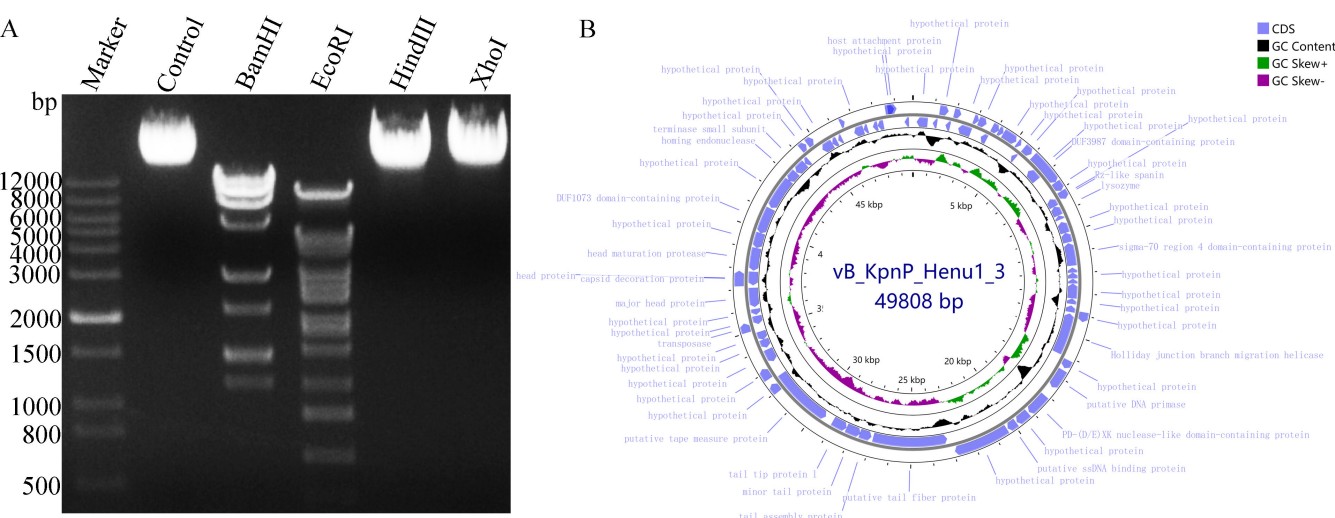

**FIG 4** Genomic DNA and comprehensive genome map of phage vB_KpnP_Henu1_3. (A) Genomic DNA of phage vB_KpnP_Henu1_3 was subjected to restriction digestion by four restriction enzymes, namely, *Bam*HI, *Eco*RI, *Hind*III, and *Xho*I. (B) Complete genome map of vB_KpnP_Henu1_3, which consists of 49,808 base pairs. A genome map of vB_KpnP_Henu1_3 was generated via the Proksee GC viewer tool. ORFs encoding all genes, annotated genes, and the GC content are depicted.

**TABLE 2** ORF analysis of the vB_KpnP_Henu1_3 phage genome

| ORF | Strand | Start | Stop | Predicted protein function | Best-match BLASTp result | Query cover | E-values | Identity | Accession | MW (kDa) |
|---|---|---|---|---|---|---|---|---|---|---|
| 1 | − | 180 | 725 | Hypothetical protein | *Klebsiella* phage RCIP0025 | 100% | 1E-130 | 99.45% | WPJ50710.1 | 20.85 |
| 2 | − | 982 | 1,200 | Putative membrane protein | *Klebsiella* phage vB_KpnS-VAC6 | 100% | 1E-43 | 97.22% | QZE50691.1 | 8.24 |
| 3 | + | 1,232 | 1,639 | Hypothetical protein | *Klebsiella* phage P528 | 100% | 6E-91 | 94.81% | QPX75233.1 | 15.34 |
| 4 | − | 1,623 | 1,856 | Hypothetical protein | No hit | –[a] | – | – | – | 8.63 |
| 5 | + | 1,903 | 2,232 | Hypothetical protein | *Klebsiella* phage RCIP0025 | 100% | 5e-72 | 97.25% | WPJ50706.1 | 12.76 |
| 6 | − | 2,273 | 2,920 | Hypothetical protein | *Klebsiella* phage KPP2020 | 53% | 6E-44 | 66.38% | WCR32885.1 | 23.86 |
| 7 | + | 2,842 | 3,081 | Hypothetical protein | *Klebsiella* phage vB_1086 | 100% | 3E-52 | 100% | UJQ43207.1 | 8.76 |
| 8 | + | 3,082 | 3,459 | Hypothetical protein | *Klebsiella* phage RCIP0078 | 100% | 5E-81 | 98.33% | WPJ54583.1 | 14.28 |
| 9 | − | 3,469 | 3,780 | Hypothetical protein | No hit | | | | | 11.59 |
| 10 | + | 3,759 | 4,100 | Hypothetical protein | *Klebsiella* phage vB_KpnS_IMGroot | 100% | 6E-78 | 99.12% | YP_009902522.1 | 12.41 |
| 11 | + | 4,093 | 4,332 | Hypothetical protein | *Bacteriophage sp.* | 100% | 9E-52 | 100% | UVN04682.1 | 8.98 |
| 12 | + | 4,340 | 5,032 | Hypothetical protein | *Klebsiella* phage vB_KpnD_Chell | 100% | 1E-170 | 98.70% | UGO54751.1 | 26.51 |
| 13 | + | 5,105 | 5,308 | Membrane protein | *Klebsiella* phage MezzoGao | 100% | 7E-39 | 100% | YP_009792140.1 | 7.48 |
| 14 | − | 5,262 | 5,480 | Hypothetical protein | *Klebsiella* phage KPP2020 | 76% | 6E-22 | 80.00% | WCR32894.1 | 8.23 |
| 15 | + | 5,489 | 5,926 | Hypothetical protein | *Klebsiella* phage vB_1086 | 100% | 2e-100 | 100% | UJQ43199.1 | 15.82 |
| 16 | + | 6,051 | 7,619 | DNA helicase | *Klebsiella* phage AloofButler | 100% | 0.0 | 99.62% | WWD15272.1 | 57.83 |
| 17 | − | 6,386 | 6,994 | Hypothetical protein | *Klebsiella* phage KPP2020 | 100% | 1e-110 | 87.13% | WCR32897.1 | 21.87 |
| 18 | + | 7,623 | 8,084 | Hypothetical protein | *Klebsiella* phage vB_LZ2044 | 100% | 1e-110 | 100% | WCF59171.1 | 17.75 |
| 19 | + | 8,096 | 8,500 | Hypothetical protein | No hit | | | | | 15.34 |
| 20 | − | 8,153 | 8,578 | Rz-like spanin | *Klebsiella* phage vB_KpnS_KpV522 | 100% | 1e-96 | 100% | YP_009787676.1 | 14.88 |
| 21 | − | 8,575 | 9,057 | Lysozyme | *Klebsiella* phage vB_1086 | 100% | 3e-115 | 99.38% | UJQ43195.1 | 17.93 |
| 22 | − | 9,059 | 9,274 | Hypothetical protein | *Klebsiella* phage vB_1086 | 100% | 4e-41 | 100% | UJQ43194.1 | 7.56 |
| 23 | − | 9,409 | 9,981 | Hypothetical protein | *Klebsiella* phage vB_LZ2044 | 100% | 1e-137 | 100% | WCF59170.1 | 21.67 |
| 24 | − | 9,978 | 10,469 | Hypothetical protein | *Klebsiella* phage RCIP0025 | 100% | 3e-115 | 99.39% | WPJ50688.1 | 18.50 |
| 25 | − | 10,508 | 11,638 | DNA repair exonuclease | *Klebsiella* phage vB_KpnS_KpV522 | 100% | 0.0 | 99.47% | YP_009787671.1 | 42.45 |
| 26 | − | 11,734 | 11,982 | Hypothetical protein | *Klebsiella* phage PKP126 | 100% | 1e-52 | 100% | YP_009284873.1 | 9.44 |
| 27 | − | 11,982 | 12,272 | Hypothetical protein | *Bacteriophage sp.* | 100% | 7e-65 | 100% | UVN04668.1 | 10.62 |
| 28 | − | 12,283 | 12,519 | Hypothetical protein | *Klebsiella pneumoniae* | 100% | 6e-50 | 100% | WP_216264657.1 | 9.07 |
| 29 | − | 12,523 | 13,254 | Hypothetical protein | *Klebsiella* phage vB_1086 | 100% | 0.0 | 99.59% | UJQ43187.1 | 27.85 |
| 30 | − | 13,257 | 13,535 | Hypothetical protein | *Klebsiella* phage vB_1086 | 100% | 4e-59 | 100% | UJQ43186.1 | 10.07 |
| 31 | − | 13,610 | 13,939 | Hypothetical protein | *Klebsiella* phage RCIP0059 | 100% | 6e-74 | 100% | WPJ53215.1 | 12.14 |
| 32 | + | 13,838 | 14,290 | Hypothetical protein | *Klebsiella* phage vB_KleS-HSE3 | 84% | 0.013 | 38.06% | QIN95008.1 | 17.04 |
| 33 | − | 14,014 | 16,050 | DNA helicase | *Klebsiella* phage vB_Kpn_K31PH164 | 100% | 0.0 | 99.85% | CAK6589201.1 | 77.10 |
| 34 | + | 16,141 | 16,542 | Hypothetical protein | *Enterobacteriaceae* | 100% | 7e-93 | 100% | WP_216264666.1 | 15.24 |
| 35 | + | 16,654 | 17,580 | Putative DNA primase | *Klebsiella* phage vB_LZ2044 | 100% | 0.0 | 100% | WCF59221.1 | 34.71 |
| 36 | + | 18,077 | 19,123 | Exonuclease | *Klebsiella* phage 209 | 100% | 0.0 | 98.56% | WKV32679.1 | 39.34 |
| 37 | + | 19,183 | 19,839 | Hypothetical protein | *Klebsiella* phage vB_1086 | 100% | 8e-161 | 100% | UJQ43180.1 | 24.34 |
| 38 | + | 19,876 | 20,343 | Putative ssDNA-binding protein | *Klebsiella* phage PWKp17 | 100% | 3e-110 | 98.71% | UJD05649.1 | 17.57 |
| 39 | + | 20,392 | 22,974 | Tail fiber protein | *Klebsiella* phage P287 | 98% | 0.0 | 98.82% | XDJ01954.1 | 91.29 |
| 40 | − | 23,198 | 26,896 | Putative tail fiber protein | *Klebsiella* phage vB_LZ2044 | 100% | 0.0 | 99.92% | WCF59234.1 | 136.69 |
| 41 | − | 26,984 | 27,586 | Tail assembly protein | *Klebsiella pneumoniae* | 100% | 2e-140 | 100% | WP_216264494.1 | 20.72 |
| 42 | − | 27,561 | 28,298 | Minor tail protein | *Klebsiella* phage vB_KpnS_IMGroot | 100% | 0.0 | 100% | YP_009902490.1 | 28.42 |
| 43 | − | 28,300 | 29,052 | Tail tip protein L | *Klebsiella* phage RCIP0040 | 100% | 0.0 | 100% | WPJ51778.1 | 27.52 |
| 44 | − | 29,469 | 32,534 | Putative tape measure protein | *Klebsiella* phage vB_LZ2044 | 100% | 0.0 | 98.92% | WCF59235.1 | 110.20 |
| 45 | + | 31,948 | 32,409 | Hypothetical protein | *Klebsiella* phage mfs | 96% | 1e-88 | 95.92% | UYE93665.1 | 15.96 |
| 46 | + | 32,668 | 33,213 | Hypothetical protein | *Klebsiella* phage KPP2020 | 33% | 8e-15 | 63.93% | WCR32835.1 | 20.60 |
| 47 | − | 33,275 | 33,931 | Hypothetical protein | *Klebsiella* phage vB_1086 | 100% | 8e-160 | 99.54% | UJQ43170.1 | 24.20 |
| 48 | − | 34,025 | 34,456 | Hypothetical protein | *Stenotrophomonas* phage vB_SmeS_BUCT705 | 100% | 8e-102 | 100% | UNY50352.1 | 15.80 |
| 49 | − | 34,446 | 34,883 | HK97 gp10 family phage protein | *Klebsiella pneumoniae* | 100% | 8e-103 | 100% | WP_216264529.1 | 16.02 |
| 50 | + | 35,014 | 35,454 | Transposase | *Klebsiella* phage KPP2020 | 100% | 3e-87 | 86.99% | WCR32841.1 | 16.33 |

(*Continued on next page*)

**TABLE 2** ORF analysis of the vB_KpnP_Henu1_3 phage genome (*Continued*)

| ORF | Strand | Start | Stop | Predicted protein function | Best-match BLASTp result | Query cover | E-values | Identity | Accession | MW (kDa) |
|---|---|---|---|---|---|---|---|---|---|---|
| 51 | − | 35,259 | 35,675 | Hypothetical protein | *Klebsiella* phage RCIP0085 | 100% | 3e-97 | 100% | WPJ55022.1 | 15.44 |
| 52 | − | 35,729 | 36,025 | Hypothetical protein | *Klebsiella* phage BUCT556A | 100% | 2e-62 | 98.98% | UPT53762.1 | 11.22 |
| 53 | − | 36,118 | 37,077 | Major head protein | *Klebsiella* phage GML-KpCol1 | 100% | 0.0 | 99.69% | YP_009796926.1 | 35.23 |
| 54 | + | 37,166 | 37,876 | Capsid decoration protein | *Klebsiella* phage vB_KpnS-VAC11 | 69% | 2e-72 | 70.91% | QZE51044.1 | 25.73 |
| 55 | − | 37,193 | 37,870 | Head protein | *Klebsiella* phage Sanco | 99% | 8e-131 | 87.50% | QBZ71160.2 | 22.93 |
| 56 | − | 37,922 | 39,052 | Putative major capsid protein | *Klebsiella* phage LAPAZ | 100% | 0.0 | 99.47% | XAG95214.1 | 41.08 |
| 57 | − | 39,049 | 39,819 | Hypothetical protein | *Klebsiella* phage vB_1086 | 100% | 0.0 | 100% | UJQ43161.1 | 29.34 |
| 58 | − | 39,809 | 41,128 | DUF1073 domain-containing protein | *Stenotrophomonas* phage vB_SmeS_BUCT705 | 100% | 0.0 | 100% | UNY50343.1 | 48.54 |
| 59 | − | 41,175 | 42,794 | Hypothetical protein | *Klebsiella* phage RCIP0025 | 98% | 0.0 | 99.81% | WPJ50655.1 | 61.96 |
| 60 | − | 42,769 | 43,320 | Homing endonuclease | *Klebsiella* phage vB_LZ2044 | 92% | 1e-122 | 99.41% | WCF59217.1 | 20.35 |
| 61 | − | 43,336 | 43,944 | Terminase small subunit | *Klebsiella* phage vB_Kpl_K53PH164C2 | 100% | 3e-144 | 97.03% | CAK6604553.1 | 22.44 |
| 62 | − | 43,941 | 44,204 | Hypothetical protein | *Klebsiella* phage PKP126 | 100% | 3e-57 | 100% | YP_009284915.1 | 9.92 |
| 63 | + | 44,240 | 44,566 | Hypothetical protein | *Escherichia coli* | 73% | 2e-50 | 100% | WP_236527818.1 | 12.31 |
| 64 | + | 44,620 | 44,967 | Hypothetical protein | No hit | | | | | 13.29 |
| 65 | − | 45,049 | 45,624 | Hypothetical protein | *Klebsiella* phage vB_1086 | 100% | 2e-139 | 98.95% | UJQ43152.1 | 22.10 |
| 66 | − | 45,621 | 45,893 | Hypothetical protein | *Klebsiella* phage vB_1086 | 100% | 4e-60 | 100% | UJQ43151.1 | 10.25 |
| 67 | − | 45,957 | 46,169 | Hypothetical protein | *Klebsiella* phage vB_1086 | 100% | 7e-45 | 98.57% | UJQ43150.1 | 8.36 |
| 68 | + | 46,390 | 46,611 | Hypothetical protein | *Escherichia coli* | 76% | 4e-26 | 89.29% | WP_236273106.1 | 8.41 |
| 69 | − | 46,831 | 47,319 | Hypothetical protein | *Klebsiella* phage vB_1086 | 100% | 2e-114 | 98.15% | UJQ43147.1 | 18.58 |
| 70 | − | 47,319 | 47,537 | Hypothetical protein | *Klebsiella* phage Shelby | 100% | 1e-41 | 97.22% | YP_009903372.1 | 8.22 |
| 71 | − | 47,799 | 48,044 | Hypothetical protein | *Klebsiella* phage KPN N141 | 100% | 2e-53 | 100% | YP_009791613.1 | 9.29 |
| 72 | − | 48,044 | 48,346 | Hypothetical protein | *Klebsiella* phage vB_1086 | 100% | 1e-67 | 99.00% | UJQ43143.1 | 11.50 |
| 73 | + | 48,539 | 48,949 | Hypothetical protein | *Klebsiella* phage KPP2020 | 90% | 1e-70 | 84.55% | WCR32871.1 | 14.92 |
| 74 | + | 48,654 | 49,064 | Host attachment protein | *Klebsiella* phage KL | 65% | 5e-08 | 42.70% | YP_009902841.1 | 15.92 |
| 75 | − | 49,390 | 49,614 | Hypothetical protein | *Klebsiella* phage vB_1086 | 100% | 1e-49 | 98.65% | UJQ43139.2 | 8.68 |

*a*−, no data.

$10^{11}$ CFU/mL). The same volume of phage vB_KpnP_Henu1_3 (MOI = 100, 10, 1, 0.1, 0.01) was used for treatment at 1 h post-infection, and the group in which the same volume of PBS was injected served as a negative control (Fig. 8A). The results revealed that all the mice in the negative control group died within 1 day, whereas the survival rates of the mice in the different MOI phage treatment groups significantly improved (Fig. 8B). At an MOI of 100, the survival rates of the mice even reached 100% (Fig. 8B). To evaluate the lytic efficiency of phage vB_KpnP_Henu1_3 against *K. pneumoniae* Kp1049, we measured the bacterial loads in various organs of the mice. The mice were injected with 100 µL of *K. pneumoniae* Kp1049 ($1 \times 10^{10}$ CFU/mL), and phage vB_KpnP_Henu1_3 (MOI = 100, 10) was used for treatment at 1 h post-infection. The organs were used for CFU counting after 24 h. The results confirmed that the CFUs/g of the heart, liver, spleen, lung, and

**TABLE 3** Comparison of vB_KpnP_Henu1_3 and homologous bacteriophages

| Phage name | Genome size | Type | Query cover of vB_KpnP_Henu1_3 | Identity of vB_KpnP_Henu1_3 | Accession |
|---|---|---|---|---|---|
| *Klebsiella* phage Henu1_3 | 49,808 bp | Circular | 100% | 100% | PQ133004 |
| *Klebsiella* phage RCIP0025 | 50,503 bp | Linear | 96% | 98.14% | OR532819.1 |
| *Klebsiella* phage vB_1086 | 49,473 bp | Linear | 93% | 98.24% | OL865411.1 |
| *Klebsiella* phage RCIP0078 | 50,552bp | Linear | 94% | 98.08% | OR532872.1 |
| *Klebsiella* phage RCIP0092 | 50,486 bp | Linear | 92% | 97.71% | OR532886.1 |
| *Klebsiella* phage RCIP0059 | 50,487 bp | Linear | 92% | 97.63% | OR532853.1 |
| *Klebsiella* phage LF20 | 50,107 bp | Linear | 86% | 96.31% | MW417503.1 |
| *Klebsiella* phage phi1_146057 | 48,907 bp | Circular | 85% | 96.23% | PP889505.1 |
| *Klebsiella* phage AloofButler | 51,584 bp | Linear | 88% | 96.15% | OR896858.1 |

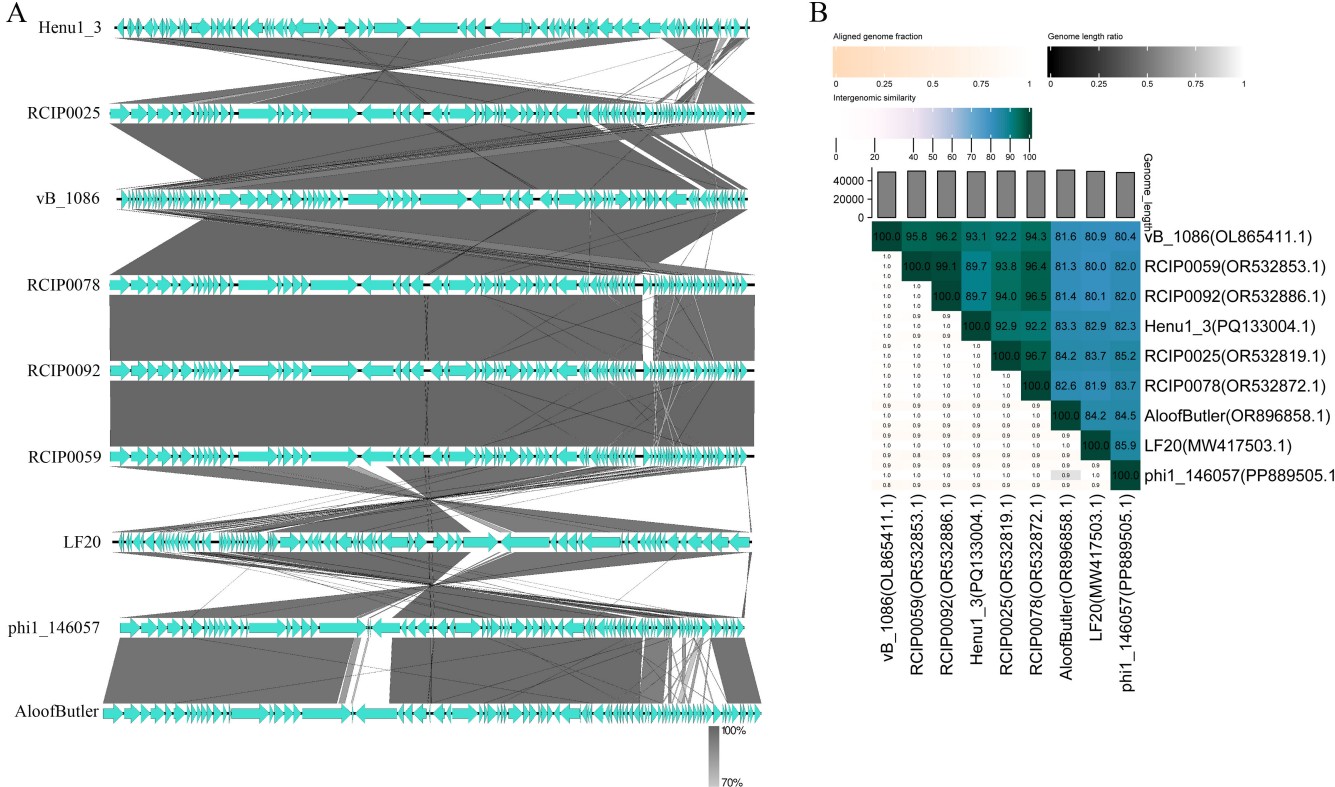

**FIG 5** Compared genomic analysis of phage vB_KpnP_Henu1_3. (A) Alignment of vB_KpnP_Henu1_3 together with similar genomes of other phages. (B) Percentage sequence similarity between the phage vB_KpnP_Henu1_3 genome and homologous phages. The values were calculated via VIDIRIC.

kidney were all prominently reduced at MOIs = 100 or 10 (Fig. 8C). These results indicated that phage vB_KpnP_Henu1_3 could effectively lyse K1-type *K. pneumoniae in vivo* and improve the survival of the mice.

## DISCUSSION

*K. pneumoniae* is an important opportunistic pathogen that can cause a variety of diseases, such as respiratory tract, circulatory system, and wound infections, and is notorious for its high mortality (39). Owing to the production of β-lactamases, the prevalence of β-lactam resistance in *K. pneumoniae* has gradually increased, and the resistance to fluoroquinolones and carbapenems has also been reported (40). In addition, *K. pneumoniae* carbapenemase (KPC) mutations, which can reduce the efficacy of ceftazidime-avibactam (CZA), the pioneer antimicrobial agent for carbapenem-resistant Enterobacteriaceae infections, have been reported in hundreds of species (41). *K. pneumoniae* can be classified into two types: classical *K. pneumoniae* (cKP) and hypervirulent *K. pneumoniae* (hvKP). cKP isolates are highly diverse and important causes of nosocomial infections. HvKP is a common pathogen that causes pyogenic liver abscesses in young healthy people and was first identified in Taiwan in 1986 (42). HvKP can infect almost all parts of the body and cause serious infections, such as meningitis, endophthalmitis, pneumonia, bacteremia, liver abscesses, and skin tissue necrosis. Compared with cKP infections, hvKP infections have a higher mortality rate, a more rapid onset of disease, and a poorer prognosis, which makes postinfectious treatment of hvKP more difficult (43). On the basis of the available statistics of *K. pneumoniae* serotypes, hvKPs include mainly K1, K2, K5, and K57, of which K1 and K2 are the most common, accounting for approximately 70% of hvKPs (44, 45). The emergence of highly resistant and hypervirulent *K. pneumoniae* poses a great threat to the clinical treatment and control of *K. pneumoniae* infections (46, 47). The biofilm formation rate of hvKP was

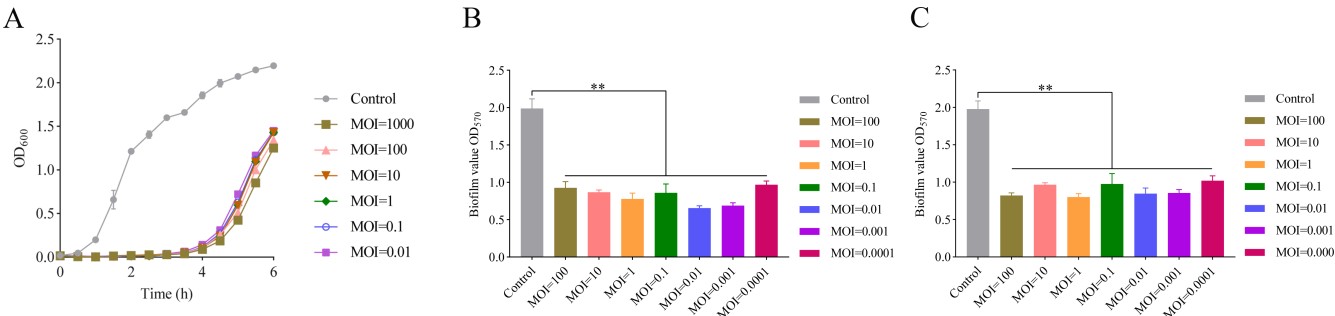

**FIG 6** Assessment of the efficacy of phage vB_KpnP_Henu1_3 against *K. pneumoniae* Kp1049 *in vitro*. (A) Inhibition of *K. pneumoniae* Kp1049 growth by phage vB_KpnP_Henu1_3. *K. pneumoniae* Kp1049 was cocultivated with vB_KpnP_Henu1_3 at different MOIs at 37°C with shaking for 6 h, and the absorbance ($OD_{600}$) of each group was measured every 0.5 h. (B) The biofilm inhibition effect of phage vB_KpnP_Henu1_3. *K. pneumoniae* kp1049 was coincubated with phage vB_KpnP_Henu1_3 at MOIs of 0.0001, 0.001, 0.01, 0.1, 1, 10, and 100 for 12 h at 37°C. After incubation, the biofilm remaining after phage vB_KpnP_Henu1_3 treatment was measured via a CV assay. (C) The biofilm disruption effect of phage vB_KpnP_Henu1_3. *K. pneumoniae* Kp1049 was incubated at 37°C for 24 h to form a mature biofilm, and then phage vB_KpnP_Henu1_3 at MOIs of 0.0001, 0.001, 0.01, 0.1, 1, 10, and 100 was coincubated with the formed biofilm for another 12 h at 37°C. The remaining biofilm was detected via a CV assay. The data are shown as the means ± SD (*n*=3). Statistical significance was analyzed by Kruskal-Wallis test, a non-parametric alternative to one-way ANOVA, followed by Dunn's post hoc test for pairwise comparisons (***$P < 0.001$, **$P < 0.01$, *$P < 0.05$).

significantly higher than that of cKP (48). The hvKP strains not only formed denser and more cohesive biofilms but also exhibited more complex extracellular matrix (48, 49). Therefore, there is an urgent need to develop alternative therapies or new methods that can be combined with antibiotic therapy to address the great threat posed by antibiotic resistance. Phages, as one of the most promising means of addressing bacterial drug resistance, are also bound to play an important role in the treatment of *K. pneumoniae* infections. The mass isolation and characterization of *Klebsiella* phages will provide selective therapeutic options for the complex capsular polysaccharide (CPS) types associated with *K. pneumoniae* infection.

Lytic phages can lyse and eliminate their target bacteria and have unique advantages, such as high specificity, efficiency, multiplication rate, and resistance prevention ability (50). These properties increase their potential for combating antibiotic-resistant bacterial infections (51). Here, we isolated and reported a novel lytic phage, vB_KpnP_Henu1_3, from hospital sewage that exhibits interesting characteristics for feasible applications in controlling *K. pneumoniae*, specifically the K1 capsule type. The phage vB_KpnP_Henu1_3 was isolated by *K. pneumoniae* Kp1049 as the host bacterium and is capable of producing phage spots with halo rings (Fig. 1A). TEM revealed that phage vB_KpnP_Henu1_3 had an ortho-icosahedral head structure and a tail length of 179.86 ± 2.01 nm (Fig. 1B). Restriction-modification (RM) systems represent a ubiquitous and evolutionarily ancient defense mechanism in bacteria and archaea, serving as a primary barrier against bacteriophage infection (52). The double-stranded DNA phage vB_KpnP_Henu1_3, while susceptible to cleavage by multiple restriction endonucleases, demonstrates resistance to *Hind*III and *Xho*I digestion (Fig. 4A). Genomic analysis reveals the absence of recognition sites for *Hind*III and *Xho*I, and no DNA modification-related proteins (e.g., methyltransferases) are encoded in its genome. These findings suggest that vB_KpnP_Henu1_3 employs alternative strategies to circumvent host RM systems, independent of widespread DNA modifications. The genome length of phage vB_KpnP_Henu1_3 is 49,808 bp, and the G + C content is 50.76%. Most of the phages reported to date are double-stranded DNA, and a few are single-stranded DNA or RNA. The genome length and G + C content of phage vB_KpnP_Henu1_3 are consistent with those of most *Klebsiella* phages. The genome sequence of phage vB_KpnP_Henu1_3 shares the highest similarity of 98.14% and 96% coverage with that of *Klebsiella* phage RCIP0025. The genome of phage vB_KpnP_Henu1_3 is circular, whereas that of *Klebsiella*

## A

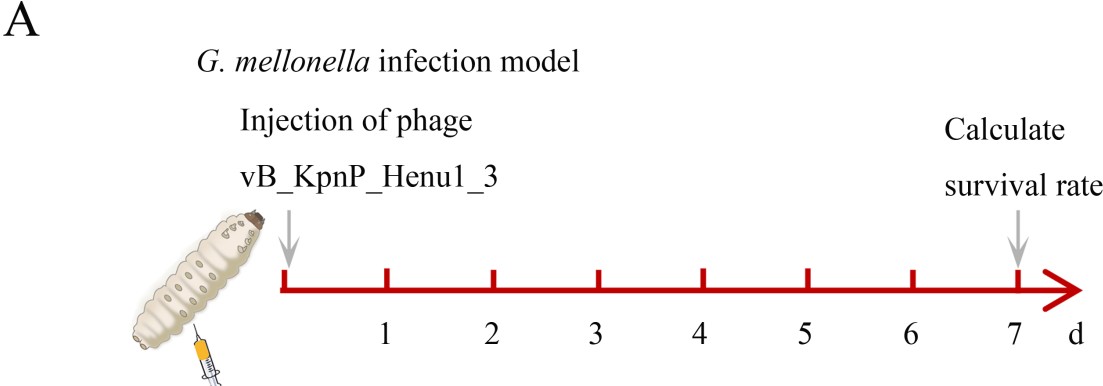

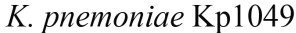

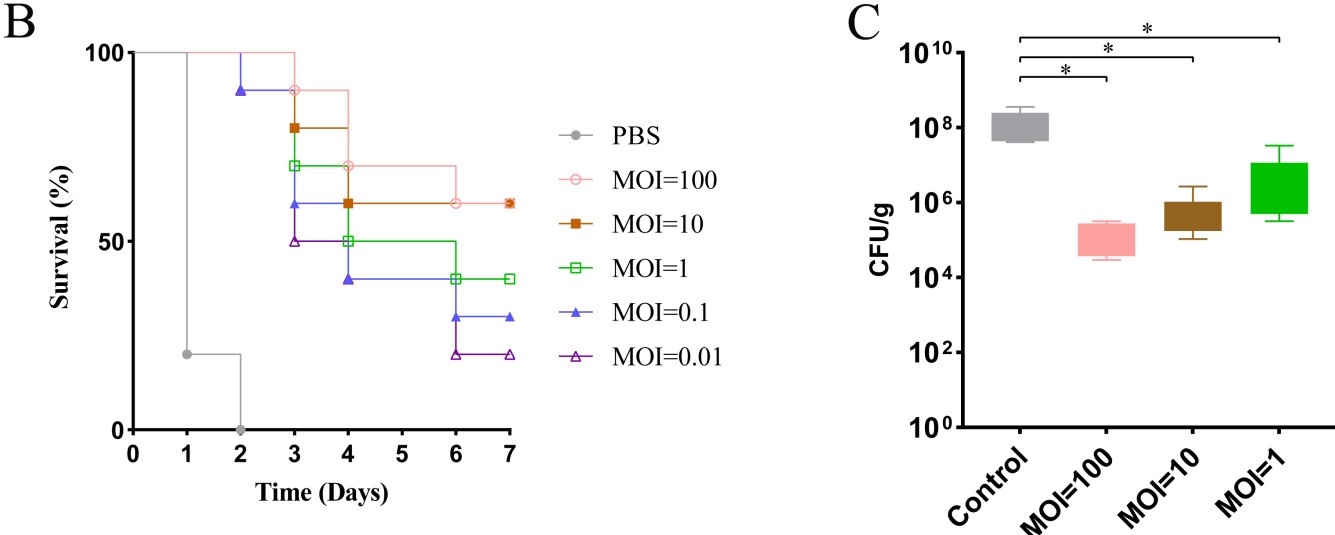

**FIG 7** Phage therapy in the *G. mellonella* larvae model. (A) Schematic representation of phage vB_KpnP_Henu1_3 treating *K. pneumoniae* Kp1049-infected *G. mellonella* larvae. (B) Survival curves of *G. mellonella* (*n* = 10) infected with *K. pneumoniae* Kp1049 followed by phage vB_KpnP_Henu1_3 treatment at different MOIs. (C) Bacterial loads in infected *G. mellonella* larvae. *G. mellonella* larvae (*n* = 6) infected with *K. pneumoniae* Kp1049 were treated with phage vB_KpnP_Henu1_3 at MOIs of 1, 10, and 100. After treatment for 24 h, the bacterial loads were determined via the serial dilution method. Statistical significance was determined by Student's *t*-test (*$P < 0.05$, vs. infected control).

phage RCIP0025 is linear. However, there are no reports describing the relationships between circular or linear DNA and phage biological characteristics.

By testing the ability of phage vB_KpnP_Henu1_3 to infect 30 clinical isolates of *K. pneumoniae*, we found that phage vB_KpnP_Henu1_3 could solely target and lyse K1-type *K. pneumoniae*. Although both the K1 and K2 types of *K. pneumoniae* are highly virulent and viscous, the phage vB_KpnP_Henu1_3 failed to lyse clinical isolates of K2-type *K. pneumoniae*, including Kp408, Kp1203, Kp1616, and Kp0524. CPS is one of the crucial virulence factors of *K. pneumoniae* and the necessary receptor of most *Klebsiella* phages (53). Differences in the structure of the *K. pneumoniae* CPS between types K1 and K2 may be the main reason for the specificity of phage vB_KpnP_Henu1_3 infection (54). Correspondingly, depolymerase recognizes and degrades the bacterial CPS, thus enabling the phage to recognize and bind to secondary receptors (usually outer membrane proteins) on the bacterial surface (55). Phage depolymerases have been identified in many bacteriophages, which recognize and degrade CPS in a host-specific

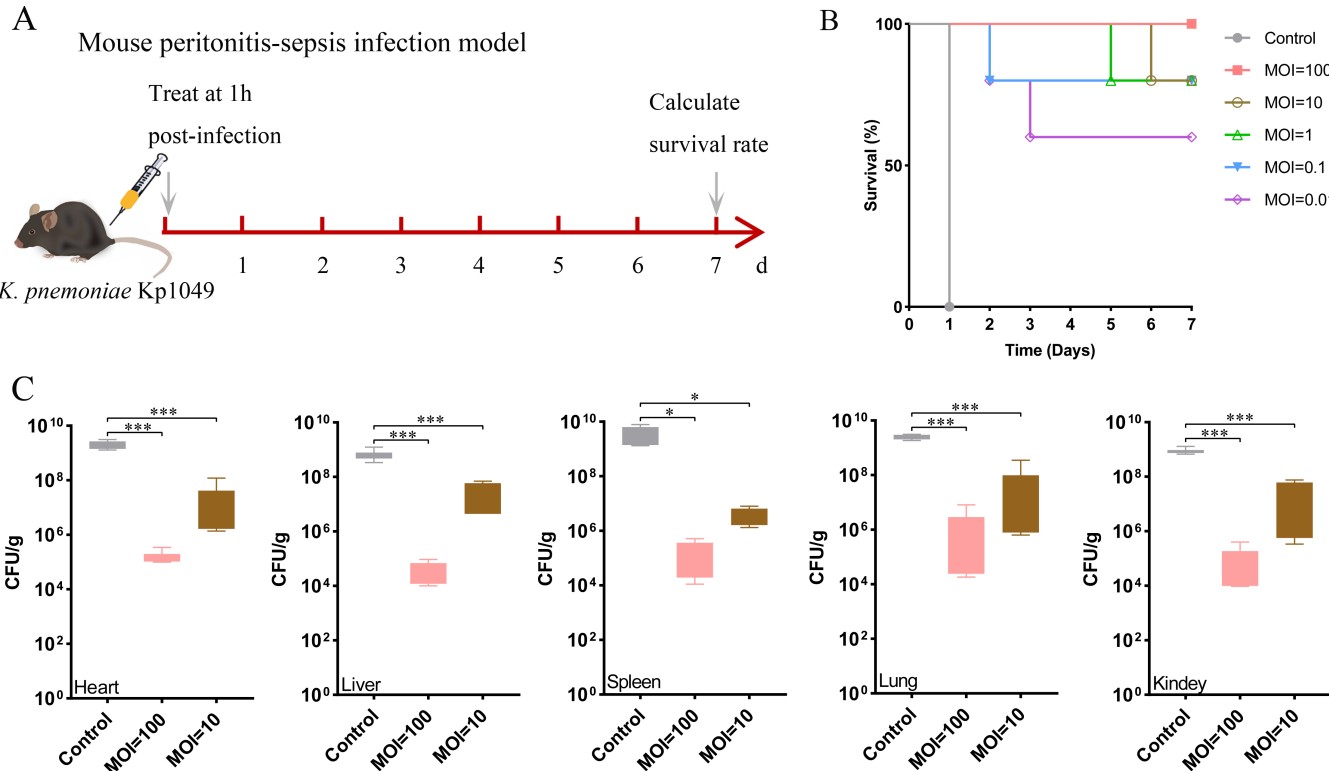

**FIG 8** Phage therapy in a mouse bacteremia model. (A) Schematic representation of phage vB_KpnP_Henu1_3 treatment of *K. pneumoniae* Kp1049-infected mice. (B) Survival rate analysis of infected mice. Mice were infected with *K. pneumoniae* Kp1049 at a CFU/mL of $1 \times 10^{11}$ and received phage therapy at different MOIs (100, 10, 1, 0.1, and 0.01) ($n = 5$). The survival of the mice was observed and recorded for 7 days. (C) Bacterial loads of different organs in a mouse bacteremia model. Mice were infected with *K. pneumoniae* Kp1049 at a concentration of $1 \times 10^{10}$ CFU/mL via intraperitoneal injection ($n = 4$). After 1 h, the infected mice were treated with vB_KpnP_Henu1_3 at different MOIs. The bacterial loads were determined by counting the CFUs in the dissected and ground tissues. Statistical significance was determined by Student's *t*-test (***$P < 0.001$, **$P < 0.01$, *$P < 0.05$, vs. untreated controls).

manner. The depolymerase Depo16 from phage ZK1 can specifically degrade the K1 serotype CPS of *K. pneumoniae* and increase the sensitivity of K1-type *K. pneumoniae* to the peritoneum macrophages (56). The depolymerase Dep42 derived from phage SH-KP152226, which is specific for the K47 capsule, was able to significantly inhibit biofilm formation or degrade formed biofilms (57). A capsule depolymerase specific for KL47-type CPS, which inhibited biofilms as well as the prevention and control of CRKP infections, was identified in the *Klebsiella* phage P560 genome (58). *Klebsiella* phage infection of fixed Capsular-type *K. pneumoniae* may be associated with phage-encoded depolymerases. In the genomes of phage vB_KpnP_Henu1_3, two ORFs are predicted to be putative tail fiber proteins that encode depolymerases (ORF39 and ORF40). Thus, these two genes may be among the main reasons for the selective infection of *K. pneumoniae* type K1 by phage vB_KpnP_Henu1_3.

The phage vB_KpnP_Henu1_3 acts in a lytic manner and can rapidly lyse K1-type *K. pneumoniae* with an optimal infection multiplicity of 0.01 (Fig. 2A). When *K. pneumoniae* Kp1049 was mixed with the phage vB_KpnP_Henu1_3, the phage was able to rapidly recognize *K. pneumoniae*, and the phage adsorbed to *K. pneumoniae* completely within 30 min (Fig. 2B). Phage vB_KpnP_Henu1_3 has a short incubation period of 20 min, and the burst size of phage vB_KpnP_Henu1_3 is approximately $253 \pm 54$ average progeny per infected cell (Fig. 2C). The lysis efficiency of phage vB_KpnP_Henu1_3 remains relatively low at 55°C, below pH 4 and above pH 11 (Fig. 3). The high burst size and lysis stability of phage vB_KpnP_Henu1_3 both set the stage for phage applications. Infection of *K. pneumoniae* Kp1049 with phage vB_KpnP_Henu1_3 at different MOIs delayed the entry of the bacteria into the log phase by approximately 4 h (Fig. 6A). However, after 4 h,

*K. pneumoniae* rapidly entered the logarithmic phase, which was probably attributed to bacterial resistance to the phage. The generation of resistance during the incubation of phages with host bacteria *in vitro* is a common phenomenon observed in many studies (59, 60). The rapid development of bacterial resistance to phages is primarily caused by receptor mutations that reduce adsorption efficiency (61). Therefore, strategies such as developing phage cocktails targeting different receptors or combining phages with antibiotics can minimize the frequency of phage resistance emergence (62, 63). These approaches will help mitigate the risk of phage therapy failure due to resistance in the future. Further confirmation of a large reservoir of novel phages is a prerequisite for the clinical application of phages. Biofilms are one of the forms of bacteria that face extreme environments and are extremely resistant to antimicrobial agents. Numerous studies have shown that phages have a clear advantage in the formation and disruption of biofilms. The phage vB_KpnP_Henu1_3 not only inhibits planktonic bacteria but also inhibits the formation of bacterial biofilms and disrupts mature biofilms (Fig. 6B and C). Genome sequencing revealed that phage vB_KpnP_Henu1_3 does not encode virulence- or resistance-related proteins. However, the functions of most of the proteins encoded by phage vB_KpnP_Henu1_3 are unknown. Therefore, further studies to reveal the functions of proteins encoded by phage vB_KpnP_Henu1_3 will lay a solid foundation for the clinical application of phages.

The ability of phages to find pathogenic bacteria and lyse them in animals is the basis for their clinical application. Currently, animal models commonly used to evaluate phage therapy for *K. pneumoniae* infections include zebrafish, *G. mellonella* larvae, and mice (38, 64–66). To further evaluate the antimicrobial effect of phage vB_KpnP_Henu1_3 *in vivo*, two animal infection models were established in this study, including *G. mellonella* larvae and mice. The results indicated that phage vB_KpnP_Henu1_3 rapidly reduced the number of bacteria in the animal infection model, and the higher the MOI was, the lower the bacterial load (Fig. 7C and 8C). Bacteria can swiftly foster phage resistance *in vitro*, while a single administration of phage vB_KpnP_Henu1_3 resulted in a significant increase in animal survival (Fig. 7B and 8B). When the MOI was 100, the survival rate of the mice reached 100% (Fig. 8B). While phage vB_KpnP_Henu1_3 demonstrated an optimal MOI of 0.01 under *in vitro* conditions, *in vivo* efficacy required significantly higher phage titers. This observation aligns with previously reported phage behavior and likely reflects the physiological challenges of systemic infections (38). The requirement for elevated phage concentrations *in vivo* may be explained by several factors: (i) extensive bacterial dissemination throughout host tissues (ii), reduced phage penetration at infection sites due to biological barriers, and (iii) the need to overcome rapid bacterial replication in host microenvironments. These physiological constraints collectively necessitate higher phage loads to ensure adequate bacterial adsorption and subsequent lysis for therapeutic efficacy.

In this study, we successfully isolated and characterized a novel lytic phage that specifically targets K1-type hvKP. We carried out morphological observation, host range analysis, biological characterization, sensitivity analysis, genomic and evolutionary characterization, and these findings revealed that this phage exhibited excellent tolerance to a broad range of pH values and a wide temperature range. The phage vB_KpnP_Henu1_3 was effective at halting biofilm formation and disrupting mature biofilms, and no genes encoding virulence-, lysogenic-, integrase-, or antibiotic resistance-related genes were found in the genome. In addition, vB_KpnP_Henu1_3 showed promising antibacterial effects *in vivo* and *in vitro*, indicating great potential as a promising alternative for antimicrobial therapy.

## ACKNOWLEDGMENTS

The present study is supported by the Natural Science Foundation of Henan (202300410052); the China Postdoctoral Science Foundation (2020M682279); Projects for College Students at Henan University (XJ2024333, S202410475020); Kaifeng Science and Technology Development Program Projects (2303007); Henan Science and Technology

Development Program Projects (242102310023, 252102310141); Henan Medical Science and Technology Research Project (LHGJ20250094); and provinces and ministries jointly building key projects of the Henan Provincial Health Commission (LHGJ20240383, SBGJ202402084).

Yuan Zhang, Jianfeng Zhang, Fang Zhou, Jiaqi Li, and Qiming Li carried out the experiments and analyzed the data. Shuai Guo, Mengzhe Liu, and Xiaoyu Shi isolated and purified the phage and analyzed its genomic information. Xinwei Zhang, Dongliang Qiao, and Lin Shi collected the clinical isolates and identified the K-type strains. Yuan Zhang and Qiming Li drafted the manuscript. Kexiao Wang, Tieshan Teng, and Youhua Yuan revised the manuscript. Yuan Zhang, Qiming Li, and Shanmei Wang conceived and designed the study, revised the manuscript, and provided research funding. All the authors have read and approved the final manuscript.

## AUTHOR AFFILIATIONS

[1]Department of Clinical Laboratory, Henan Provincial People's Hospital, People's Hospital of Zhengzhou University, and People's Hospital of Henan University, Zhengzhou, China
[2]Henan Province Engineering Technology Research Center of Rapid-Accuracy Medical Diagnostics, Department of Clinical Laboratory, The First Affiliated Hospital of Henan University, Henan University, Kaifeng, China
[3]Department of Microbiology, College of Basic Medical Sciences, Henan University, Kaifeng, China
[4]The Jointed National Laboratory of Antibody Drug Engineering, Henan University, Kaifeng, China

## AUTHOR ORCIDs

Yuan Zhang  https://orcid.org/0000-0002-9297-8932
Qiming Li  http://orcid.org/0000-0002-6916-3954
Shanmei Wang  http://orcid.org/0009-0000-3439-6099

## FUNDING

| Funder | Grant(s) | Author(s) |
| --- | --- | --- |
| Natural Science Foundation of Henan Province | 202300410052 | Qiming Li |
| China Postdoctoral Science Foundation | 2020M682279 | Qiming Li |
| Projects for College Students at Henan University | XJ2024333, S202410475020 | Qiming Li |
| Kaifeng Science and Technology Development Program Projects | 2303007 | Qiming Li |
| Henan Science and Technology Development Program Projects | 242102310023 | Qiming Li |
| Henan Science and Technology Development Program Projects | 252102310141 | Yuan Zhang |
| Provinces and ministries jointly building key projects of Henan Provincial Health Commission | LHGJ20240383 | Xinwei Zhang |
| Provinces and ministries jointly building key projects of Henan Provincial Health Commission | SBGJ202402084 | Qiming Li |
| Henan Medical Science and Technology Research Project | LHGJ20250094 | Yuan Zhang |

## AUTHOR CONTRIBUTIONS

Yuan Zhang, Writing – original draft, Writing – review and editing | Fang Zhou, Data curation | Kexiao Wang, Writing – original draft, Writing – review and editing | Qiming Li, Data curation, Writing – original draft, Writing – review and editing.

## DATA AVAILABILITY

The genome sequence and annotation of phage vB_KpnP_Henu1_3 have been submitted to the GenBank database under accession number PQ133004.1. The raw sequencing data have been deposited in the National Microbiology Data Center under BioProject accession number NMDC10020201.

## ETHICS APPROVAL

Ethics approval for the study was obtained from Henan University Ethics Committee and ethics number HUSOM2025-674 was assigned to the study. All procedures were conducted following the relevant guidelines.

## ADDITIONAL FILES

The following material is available online.

### Supplemental Material

**Tables S1 (Spectrum00931-25-s0001.doc).** The isolation information of clinical isolates used in this study.

### Open Peer Review

**PEER REVIEW HISTORY (review-history.pdf).** An accounting of the reviewer comments and feedback.

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
