## [Reviewer comments · Microbiology Spectrum]

Microbiology Spectrum

Characterization of the bacteriophage vB_KpnP_Henu1_3 lytic for K1 *Klebsiella pneumoniae* and its therapeutic efficacy in *Galleria mellonella* larvae and mice

Yuan Zhang, Lin Shi, fang zhou, Jiaqi Li, Mengzhe Liu, Shuai Guo, Xiaoyu Shi, Xin-Wei Zhang, kexiao wang, Tieshan Teng, Youhua Yuan, qiming li, Shanmei Wang, Jiangfeng Zhang, and Dongliang Qiao

Corresponding Author(s): Yuan Zhang, Henan Provincial People's Hospital

Review Timeline:

Submission Date:	March 27, 2025
Editorial Decision:	May 23, 2025
Revision Received:	July 12, 2025
Editorial Decision:	July 29, 2025
Revision Received:	September 10, 2025
Editorial Decision:	October 11, 2025
Revision Received:	November 5, 2025
Editorial Decision:	November 16, 2025
Revision Received:	November 18, 2025
Accepted:	December 5, 2025

Editor: Leiliang Zhang

Reviewer(s): Disclosure of reviewer identity is with reference to reviewer comments included in decision letter(s). The following individuals involved in review of your submission have agreed to reveal their identity: Ramzi Atiah Alahmadi (Reviewer #2)

Transaction Report:

DOI: <https://doi.org/10.1128/spectrum.00931-25>

Re: Spectrum00931-25 (**Characterization of the bacteriophage vB_KpnP_Henu1_3 lytic for K1 *Klebsiella pneumoniae* and its therapeutic efficacy in *Galleria mellonella* larvae and mice**)

Dear Dr. Yuan Zhang:

Thank you for the privilege of reviewing your work. Below you will find my comments, instructions from the Spectrum editorial office, and the reviewer comments.

Revision Guidelines

Sincerely,
Leiliang Zhang
Editor
Microbiology Spectrum

Reviewer #1 (Comments for the Author):

Zhang et al. report the isolation and characterization of a novel lytic bacteriophage, vB_KpnP_Henu1_3, that specifically targets capsular type K1 *Klebsiella pneumoniae*. The authors demonstrate the phage's thermal and pH stability, its ability to inhibit and eradicate biofilms in vitro, and its effectiveness in improving survival and reducing bacterial burden in in vivo models. While the manuscript covers a wide array of methodologies, transitioning from in vitro to in vivo characterization, it would benefit from the

inclusion of several crucial methodological details and clarifications to support the robustness and reproducibility of the study.

Major methodological concerns:

1. The manuscript lacks essential information regarding the isolation and purification of the phage. This is a critical step that must be described in sufficient detail.
2. There is no description of the host range methods, despite including the results of the assays.
3. The adsorption curve assay is not described, even though it is part of the results section.
4. Statistical analyses are not described anywhere in the manuscript (not in Materials and Methods section, nor in Figure legends); details of the methods used and statistical significance thresholds should be included.
5. The TEM preparation protocol is not described-sample preparation (fixation, staining) for transmission electron microscopy should be detailed.
6. There is no description of how pH buffers were adjusted during stability assays.
7. The DNA extraction protocol, though cited, should be briefly summarized for completeness.
8. The authors should specify the cutoff value for low-quality reads during sequencing quality control.
9. The biofilm inhibition and eradication protocols are insufficiently described-concentrations, treatment times, and methods used for staining and quantification need clarification.
10. The biofilm-forming capacity of the strain is not defined-was it strongly or moderately forming? What cutoff values were used?
11. The calculation of burst size should be clarified-how was the progeny number determined? The results from the one-step growth curve seem inconsistent with a burst size of 297
12. The manuscript lacks an explanation of how circularity of the genome was determined.

Recommended additional experiments:

- Efficiency of plating (EOP) assays are recommended to quantify host range effectiveness, particularly for clinical applications. It can provide valuable information on the ability of the phage to lyse clinical strains
- A comparative genomic analysis of ORFs 39 and 40 (putative depolymerases) versus Depo16, a known depolymerase active against K1-type capsules, is suggested to confirm the author's hypothesis.

Protocol-specific questions and inconsistencies:

- Lines 122-127: The MOI optimization protocol is unclear and should be re-explained in detail.
- Line 134: Were all phage titrations done only after 2 hours, or every 10 minutes, as is standard in one-step growth curves?
- Line 175: Were the MOIs for biofilm experiments the same as for planktonic assays?
- Lines 178-191: Why were different MOIs used for survival vs. bacterial load assays? Why was 24 h selected as the endpoint for euthanizing animals?
- Lines 193-209: The mouse model section is confusing-information is lacking on the age of mice, ethics approval number, and the overseeing institutional committee.
- Lines 213-214: It is highly unlikely that only one plaque morphology was detected. Plaques should be purified 3-5 times to ensure homogeneity.
- Table 1: Year of isolation of tested strains is missing. Is there antibiotic resistance profile data for these strains?
- Lines 269-272: These sentences are grammatically incorrect and should be revised.

Comments on results and interpretations:

- Lines 217-218: The phage lifestyle cannot be determined from plaque and virion morphology. Sequencing data should be used to rule out lysogeny-associated genes.
- Lines 256-257: What was the purpose of the restriction enzyme digestion? The results are neither discussed nor contextualized.

o Note: Restriction enzyme digestion patterns, together with sequencing data, may offer insights into methylation patterns, which can influence host restriction-modification systems and phage infectivity.

- The genomic assignment to a new species within Webervirus genus (as claimed in the abstract) is not supported in the results. Whole-genome analysis and ANI/AF calculations should be shown.
- Line 311-312: The conclusion is overstated. While the phage reduces bacterial burden, it does not completely eliminate it.
- Line 317: Was it established beforehand that animals develop sepsis 1 hour after infection with the bacterial dose used? How was this timeline selected?

- Figure 5D: The phage characterized in this study should be clearly indicated, either by bolding, highlighting, or using a specific symbol to visually distinguish it from others.

Writing and grammatical corrections:

- The introduction feels disorganized-engineered phages are discussed before naturally occurring ones.
- Results should be written in present tense.
- Lines 150-151 (GC content): This information belongs in the results, not materials and methods.
- Line 51-52: The sentence is grammatically incorrect and needs rewriting.
- Line 56: Replace "ultrabroad" with "extended".
- Lines 71-72: The claim that phages are routinely used prior to antibiotics is misleading.
- Line 73: Use "increase" instead of "increased".
- Line 166: Likely an error where the phage is compared to itself. Clarify which phage the authors refer to.
- Lines 171-172, 236: It is the bacterium that was infected, not the phage.
- Line 404: The phage is likely lytic, not lysogenic.
- Line 411: The sentence on pH is inverted-lysis efficiency decreases below pH 4 and above pH 11.
- Line 433: Remove repetition of "were established in this study."

- Line 435: Remove "was".
- Lines 444-446: The sentence needs grammar revision.

Reviewer #2 (Comments for the Author):

- In materials and methods section:

Although the number of mice used for the survival analysis (n=5 per group) is acceptable for preliminary studies, the use of only four mice per group (n=4) for bacterial load analysis is a limitation. This small sample size may reduce the statistical power and increase variability. A larger group size (n{greater than or equal to}6) is recommended to strengthen the reliability and reproducibility of the results.

- In discussion section:

1- The phage targets only K1 and not K2, but it did not sufficiently explain why the phage showed no activity against K2, despite the similarity in virulence between the two types.

2- Absence of explanation for bacterial regrowth after 4 hours: It was noted that resistance appeared after 4 hours of phage treatment, but the mechanism of resistance or possible solutions were not sufficiently discussed.

- In result:

The phage showed strong lytic activity, stability across a wide range of temperatures and pH levels, and no harmful genes in its genome. It effectively reduced bacterial growth and biofilms in laboratory tests, and significantly improved survival in both insect and mouse infection models, suggesting its potential as a safe and effective treatment for infections caused by multidrug-resistant *K. pneumoniae*.

- The discussion:

provides a solid biological and contextual basis for supporting the use of phage therapy against hypervirulent *K. pneumoniae*, effectively linking the findings to previous studies, especially in terms of host specificity and the role of depolymerases. However, the discussion could be strengthened by more clearly distinguishing between lytic and lysogenic phage behavior, avoiding repetition through more concise wording, and shedding light on unexplored genomic aspects such as hypothetical proteins. Additionally, a more detailed discussion of the differences in multiplicity of infection (MOI) effectiveness between laboratory and in vivo settings would enhance the practical value of the study.

Manuscript ID: Spectrum00931-25

Manuscript title: Characterization of the bacteriophage vB_KpnP_Henu1_3 lytic for K1 *Klebsiella pneumoniae* and its therapeutic efficacy in *Galleria mellonella* larvae and mice

Dear editor,

Thank you for your thoughtful review of our manuscript and the helpful comments from the reviewers. We are pleased to hear that the manuscript will be potentially acceptable for publication. We agree with the reviewers' suggestions and have revised the manuscript accordingly for your reconsideration. Here below a point-by-point reply to the comments. Hopefully you will find the revised manuscript improved and suitable for publication on *Microbiology Spectrum*.

Yours,
Shanmei Wang

Reviewer #1

-Major methodological concerns:

Q1. The manuscript lacks essential information regarding the isolation and purification of the phage. This is a critical step that must be described in sufficient detail.

Thank you very much for raising this question. Indeed, the description of this method is crucial for a phage-related paper. We have now supplemented the phage isolation and purification protocol, which has been included as the first section of the 'Materials and Methods'.

Q2. There is no description of the host range methods, despite including the results of the assays.

Thank you very much. As suggested, we have incorporated additional experimental methods related to host range determination in the 'Materials and Methods' section.

Q3. The adsorption curve assay is not described, even though it is part of the results section.

Thank you very much. As suggested, we have incorporated additional experimental methods related to adsorption curve assay in the 'Materials and Methods' section.

Q4. Statistical analyses are not described anywhere in the manuscript (not in Materials and Methods section, nor in Figure legends); details of the methods used and statistical significance thresholds should be included.

We sincerely appreciate your suggestion. Accordingly, we have expanded the 'Materials and Methods' section to include detailed protocols for statistical analysis methods and their corresponding threshold.

Q5. The TEM preparation protocol is not described-sample preparation (fixation, staining) for transmission electron microscopy should be detailed.

Following your suggestion, we have now included detailed procedures for transmission electron microscopy (TEM) analysis in the revised manuscript.

Q6. There is no description of how pH buffers were adjusted during stability assays.

Following your suggestion, we have now included detailed procedures for pH stability of phages.

Q7. The DNA extraction protocol, though cited, should be briefly summarized for completeness.

Thank you for your suggestions! We have refined the DNA extraction protocol for phage vB_KpnP_Henu1_3 as follows: (1) Host nucleic acid removal: Purified phage was treated with DNase I and RNase A to eliminate host-derived DNA/RNA contamination. (2) DNA release: EDTA, proteinase K, and SDS were added to lyse the nucleocapsid and liberate genomic DNA. (3) DNA purification: Proteins were removed by Tris-phenol/chloroform extraction, followed by ethanol precipitation. (4) DNA dissolution: The pellet was resuspended in sterile distilled water for subsequent high-throughput sequencing.

Q8. The authors should specify the cutoff value for low-quality reads during sequencing quality control.

Thank you for your suggestions! Paired-end reads were discarded if either read in the pair had a length ≤ 50 bp.

Q9. The biofilm inhibition and eradication protocols are insufficiently described-concentrations, treatment times, and methods used for staining and quantification need clarification.

According to your suggestions, we have provided detailed descriptions of the phage-mediated biofilm inhibition and eradication assays in the Materials and Methods section.

Q10. The biofilm-forming capacity of the strain is not defined-was it strongly or moderately forming? What cutoff values were used?

Thanks for your question. The biofilm formation rate of hvKP was significantly higher than that of cKP. The hvKP strains not only formed denser and more cohesive biofilms but also exhibited more complex extracellular matrix. *K. pneumoniae* Kp1049 is a K1-type strain belonging to hvKP, which exhibits strong intrinsic biofilm-forming capacity. Phage treatment significantly inhibited biofilm formation by Kp1049. The assessment of biofilm formation ability was performed according to the methodology described in the following references.

Wen Z, Chen Y, Liu T, Han J, Jiang Y, Zhang K. 2024. Predicting Antibiotic Tolerance in hvKP and cKP Respiratory Infections Through Biofilm Formation Analysis and Its Resistance Implications. *Infect Drug Resist* 17:1529-1537.

Taha MS, Elkolaly RM, Elhendawy M, Elatrozy H, Amer AF, Helal R, Salem H, El Feky YG, Harkan A, Mashaal RG, Allam AA, Oraiby AE, Abdeen NSM, Bahey MG. 2024. Phenotypic and Genotypic Detection of Hypervirulent *Klebsiella pneumoniae* Isolated from Hospital-Acquired Infections. *Microorganisms* 12.

Q11. The calculation of burst size should be clarified-how was the progeny number determined? The results from the one-step growth curve seem inconsistent with a burst size of 297

We sincerely appreciate your inquiry. In response to your question, we meticulously re-examined the raw data and subsequently repeated the one-step growth curve assay in strict accordance with the established protocol cited in the literature. The updated results have now been incorporated into the manuscript. The burst size= $2.94 \times 10^{10} / 1 \times 10^8$

Q12. The manuscript lacks an explanation of how circularity of the genome was determined.

After sequencing the DNA of the bacteriophage, it was assembled and pieced together, revealing that the bacteriophage's genome is circular.

- *Recommended additional experiments:*

Q13. Efficiency of plating (EOP) assays are recommended to quantify host range effectiveness, particularly for clinical applications. It can provide valuable information on the ability of the phage to lyse clinical strains

As per your suggestion, we introduced the EOP (Efficiency of Plating) to evaluate the lytic efficiency of the bacteriophage against strains within its host range.

Q14. A comparative genomic analysis of ORFs 39 and 40 (putative depolymerases) versus Depo16, a known depolymerase active against K1-type capsules, is suggested to confirm the author's hypothesis.

According to your suggestion, we performed multiple sequence alignment of ORF39 and ORF40 with Depo16 and found that the similarity was not high. Later, we will confirm the polysaccharide depolymerase encoded by vB_KpnP_Henu1_3 through protein purification and expression methods.

- Protocol-specific questions and inconsistencies:

Q15. Lines 122-127: The MOI optimization protocol is unclear and should be re-explained in detail.

Based on your suggestions, we have optimized this section of the experimental methodology and provided more detailed descriptions of the experimental procedures.

Q16. Line 134: Were all phage titrations done only after 2 hours, or every 10 minutes, as is standard in one-step growth curves?

In the one-step growth curve experiment, phage titers were measured at 10-minute intervals over a 120-minute period. We have provided comprehensive details of the experimental procedures in the methodology section.

Q17. Line 175: Were the MOIs for biofilm experiments the same as for planktonic assays?

In the biofilm inhibition assay, the quantity of infecting phages was determined based on the initial bacterial inoculum. For the mature biofilm disruption experiment, the phage infection dose was instead determined according to the bacterial load within the established biofilm. We have provided a comprehensive description of the experimental methodology for this section.

Q18. Lines 178-191: Why were different MOIs used for survival vs. bacterial load assays?

Why was 24 h selected as the endpoint for euthanizing animals?

In the animal experiments, we initially observed the survival rates of infected mice and found that higher phage titer treatments yielded better therapeutic outcomes. Consequently, for the subsequent bacterial load assessment in mouse organs, we selected the MOI that demonstrated optimal treatment efficacy. This approach allowed us to evaluate the capability of phage vB_KpnP_Henu1_3 to rapidly reduce bacterial burdens. The 24-hour endpoint for mouse euthanasia was determined based on a comprehensive review of published literature and our preliminary experimental data. This time frame was selected because shorter or longer durations would compromise our ability to observe the therapeutic effects of phage treatment.

Q19. Lines 193-209: The mouse model section is confusing-information is lacking on the

age of mice, ethics approval number, and the overseeing institutional committee.

According to your suggestions, we have supplemented the detailed information of the mice and added the approval number from the ethics committee for the experiments we conducted. The ethics committee approval number has been placed at the end of the main text.

Q20 Lines 213-214: It is highly unlikely that only one plaque morphology was detected.

Plaques should be purified 3-5 times to ensure homogeneity.

Thank you for your question. We have added the process of phage isolation and purification in the Materials and Methods section. The description here should refer to the plaque morphology after five rounds of purification, which has been detailed in the manuscript.

Q21 Table 1: Year of isolation of tested strains is missing. Is there antibiotic resistance profile data for these strains?

Per your recommendations, we have now supplemented the bacterial strains with their isolation dates and detailed their antibiotic resistance profiles in the Materials and Methods section.

Q22 Lines 269-272: These sentences are grammatically incorrect and should be revised.

Based on your suggestions, we have made the corresponding revisions.

-Comments on results and interpretations:

Q23 Lines 217-218: The phage lifestyle cannot be determined from plaque and virion morphology. Sequencing data should be used to rule out lysogeny-associated genes.

Thank you for your valuable question. We have re-evaluated the phage genome using online bioinformatics tools and confirmed the absence of lysogeny-related genes, which clearly indicates that this is a strictly lytic phage. Regarding the morphological characterization results, we agree that describing it as a "virulent phage" was not entirely accurate, and we have accordingly revised our conclusions.

Q24 Lines 256-257: What was the purpose of the restriction enzyme digestion? The results are neither discussed nor contextualized.

o Note: Restriction enzyme digestion patterns, together with sequencing data, may offer insights into methylation patterns, which can influence host restriction-modification systems and phage infectivity.

In accordance with your suggestions, we have incorporated the analysis and interpretation of these results into the Discussion section.

Q25 The genomic assignment to a new species within Webervirus genus (as claimed in

the abstract) is not supported in the results. Whole-genome analysis and ANI/AF calculations should be shown.

In line with your suggestions, the taxonomic position of bacteriophage Henu1_3 has been reclassified according to the most recent ICTV guidelines for viral taxonomy.

Q26 Line 311-312: The conclusion is overstated. While the phage reduces bacterial burden, it does not completely eliminate it.

Thank you for your question. We have revised the sentence to:

"The above results demonstrated that phage vB_KpnP_Henu1_3 significantly reduced the bacterial load of *K. pneumoniae* Kp1049 in infected *G. mellonella* larvae within 24 h."

Q27 Line 317: Was it established beforehand that animals develop sepsis 1 hour after infection with the bacterial dose used? How was this timeline selected?

The mouse infection model was established based on published literature. Intraperitoneal injection of a lethal dose of bacteria was used to induce fatal septicemia in untreated mice. However, phage therapy rapidly reduced bacterial load, thereby preventing the development of sepsis and improving survival.

Zhao D, Tang M, Ma Z, Hu P, Fu Q, Yao Z, Zhou C, Zhou T, Cao J. Synergy of bacteriophage depolymerase with host immunity rescues sepsis mice infected with hypervirulent *Klebsiella pneumoniae* of capsule type K2. *Virulence*. 2024 Dec;15(1):2415945. doi: 10.1080/21505594.2024.2415945.

Q28 Figure 5D: The phage characterized in this study should be clearly indicated, either by bolding, highlighting, or using a specific symbol to visually distinguish it from others.

Thank you for your question. We have marked the location of phage vB_KpnP_Henu1_3 with a red asterisk in Figure 5C and D.

-Writing and grammatical corrections:

Q29 The introduction feels disorganized-engineered phages are discussed before naturally occurring ones.

According to your suggestions, we have revised this section of the content.

Q30 Results should be written in present tense.

Based on your suggestions, we have revised the results section using the present tense.

Q31 Lines 150-151 (GC content): This information belongs in the results, not materials

and methods.

Thanks for your comments. We have removed this statement from Materials and Methods and present it in the Results section.

Q32 Line 51-52: The sentence is grammatically incorrect and needs rewriting.

Based on your suggestions, the ambiguous descriptions have been revised accordingly.

Q33 Line 56: Replace "ultrabroad" with "extended".

Thanks for your comments. We have modified "ultrabroad" to "extended".

Q34 Lines 71-72: The claim that phages are routinely used prior to antibiotics is misleading.

Thanks for your comments. We have removed the statement in question .

Q35 Line 73: Use "increase" instead of "increased".

Thanks for your comments. We have modified "increase" to "increased".

Q36 Line 166: Likely an error where the phage is compared to itself. Clarify which phage the authors refer to.

The term "phage sequences" here specifically refers to those identified through BLASTn searches as exhibiting similarity to Henu1_3. We have revised the manuscript to clarify this previously ambiguous description.

Q37 Lines 171-172, 236: It is the bacterium that was infected, not the phage.

Thanks for your comments. We have revised the relevant statements in accordance with the recommendations.

Q38 Line 404: The phage is likely lytic, not lysogenic.

Thanks for your comments. We have modified "lysogenic" to "lytic".

Q39 Line 411: The sentence on pH is inverted-lysis efficiency decreases below pH 4 and above pH 11.

Thanks for your comments. We have modified the statement according to your suggestion.

Q40 Line 433: Remove repetition of "were established in this study."

Thanks for your comments. We have removed the duplicates.

Q41 Line 435: Remove "was".

Thanks for your comments. We have deleted "was".

Q42 Lines 444-446: The sentence needs grammar revision.

Thanks for your comments. We have revised the relevant sections.

Reviewer #2

In materials and methods section:

Q1. Although the number of mice used for the survival analysis (n=5 per group) is acceptable for preliminary studies, the use of only four mice per group (n=4) for bacterial load analysis is a limitation. This small sample size may reduce the statistical power and increase variability. A larger group size (n{greater than or equal to}6) is recommended to strengthen the reliability and reproducibility of the results.

We sincerely appreciate your valuable suggestion. In response, we have increased the number of mice per group (n = 6) in the bacterial load experiments to enhance the reproducibility and statistical power of our data. This modification has been implemented throughout the relevant experimental sections.

- In discussion section:

Q2- The phage targets only K1 and not K2, but it did not sufficiently explain why the phage showed no activity against K2, despite the similarity in virulence between the two types.

Thanks for your comments. The K1 and K2 types of *Klebsiella pneumoniae* exhibit distinct differences in their capsular polysaccharide structures. Since the capsular polysaccharide serves as the primary receptor for most phages, the inability of phage vB_KpnP_Henu1_3 to infect K2-type *K. pneumoniae* may be attributed to its failure to recognize the K2 capsular polysaccharide. We have further elaborated on this possible reason for the infection discrepancy in the Discussion section and included appropriate references to support this explanation.

Q3- Absence of explanation for bacterial regrowth after 4 hours: It was noted that resistance appeared after 4 hours of phage treatment, but the mechanism of resistance or possible solutions were not sufficiently discussed.

We appreciate your insightful question regarding the phage tolerance observed after 4 hours of treatment. This phenomenon may result from genetic mutations leading to abnormal receptor expression in bacterial hosts. As suggested, we have supplemented the discussion section with potential strategies to address phage tolerance, including phage cocktail formulation, receptor modification inhibitors or combination of antibiotics and phages.

- In result:

Q4. The phage showed strong lytic activity, stability across a wide range of temperatures and pH levels, and no harmful genes in its genome. It effectively reduced bacterial growth and biofilms in laboratory tests, and significantly improved survival in both insect and mouse infection models, suggesting its potential as a safe and effective treatment for infections caused by multidrug-resistant *K. pneumoniae*.

Thank you for your comments. We have thoroughly reviewed the Results section and made comprehensive corrections to the identified errors and inaccuracies.

- The discussion:

Q5. provides a solid biological and contextual basis for supporting the use of phage therapy against hypervirulent *K. pneumoniae*, effectively linking the findings to previous studies, especially in terms of host specificity and the role of depolymerases. However, the discussion could be strengthened by more clearly distinguishing between lytic and lysogenic phage behavior, avoiding repetition through more concise wording, and shedding light on unexplored genomic aspects such as hypothetical proteins. Additionally, a more detailed discussion of the differences in multiplicity of infection (MOI) effectiveness between laboratory and *in vivo* settings would enhance the practical value of the study.

Thank you for your comments. In the main text, we have provided a detailed discussion on the host specificity of phage vB_KpnP_Henu1_3, including its potentially encoded polysaccharide depolymerases and putative proteins within its genome. We have also thoroughly explained the reasons for the differences in the optimal MOI between *in vitro* and *in vivo* phage infection conditions. We believe that incorporating your suggested revisions will significantly improve the accessibility and acceptance of this article among readers.

Re: Spectrum00931-25R1 (**Characterization of the bacteriophage vB_KpnP_Henu1_3 lytic for K1 *Klebsiella pneumoniae* and its therapeutic efficacy in *Galleria mellonella* larvae and mice**)

Dear Dr. Yuan Zhang:

Thank you for the privilege of reviewing your work. Below you will find my comments, instructions from the Spectrum editorial office, and the reviewer comments.

Revision Guidelines

Sincerely,
Leiliang Zhang
Editor
Microbiology Spectrum

Reviewer #1 (Comments for the Author):

I appreciate the thorough efforts made by the authors in addressing the reviewers' concerns and revising the manuscript accordingly. The revised version of the manuscript demonstrates significant improvements, especially in methodological transparency and clarification of the in vivo work. The inclusion of detailed protocols for phage isolation, host range determination, TEM, adsorption curves, biofilm assays, has greatly enhanced the reproducibility and rigor of the study. The

authors have also adequately responded to the call for expanded discussion around biofilm formation, phage resistance mechanisms, and therapeutic efficacy in animal models.

However, some important issues remain that were either only partially addressed or not fully resolved:

1. Phylogenetic Support for Taxonomy:

While the authors revised the phage's taxonomic assignment, the phylogenetic analysis presented to support this placement is somewhat unclear. Since the manuscript does not specify any information on the sequences included in the phylogenetic trees generated, nor does it mention or highlight which sequences belong to which genus, it is difficult to conclude about the taxonomy. In addition, and very importantly, Siphoviridae is no longer a viral family since the new ICTV classification. A phage can have a siphovirus morphology and be classified within the Caudoviricetes class, but the family to which it belongs is different and should be correctly mentioned.

2. Statistical analysis:

Despite a new section having been included in the Materials and Methods section, there is no mention of the statistical tests used, neither in this section nor in the figure legends. This is a must that needs to be included.

3. Burst Size Calculation and Consistency:

The reported burst size ($2.94 \times 10^{10} / 1 \times 10^8 = \sim 294$) is inconsistent with the growth curve. Although the authors repeated the experiment, it is still unclear how they determined the number of infected cells at $t=0$, and whether appropriate dilutions and plaque counts were performed to ensure accurate MOI and burst size estimation. The description could benefit from a clearer step-by-step explanation. Even though the initial bacterial culture had a titer of 1×10^8 CFU/ml, the experiment was done at a 0.01 MOI, implying that approximately 1×10^6 CFU/ml were infected at $t=0$. Therefore, considering the burst size calculation (titer of progeny phage = 2.94×10^{10} divided by the titer of infected bacteria = 1×10^6), the burst size should be 2.94×10^4 phages/infected bacteria. This is also in accordance with the one-step growth curve from Fig. 2C, since another formula for calculating the burst size is: PFU/ml at plateau divided by PFU/ml at latent period. Please revise.

4. EOP Results Description:

While the authors state that EOP testing was conducted and it has indeed been added to the table, there is no mention of it in Materials and Methods nor the results section.

5. Genomic Evidence for Lytic Lifestyle:

The response indicates that lysogeny-related genes were ruled out using online tools, but this analysis is not shown or referenced in the manuscript itself. A brief mention of which tools (e.g., PhageAI, PHACTS, or manual curation of integrase/repressor genes) were used would strengthen this conclusion.

6. Supplementary tables needed:

The detailed information on the bacterial strains, currently included in the revised Materials and Methods section, would be more appropriately presented as supplementary data. This should include, at minimum: strain designations, dates and sources of isolation, antibiotic susceptibility profiles, and, where available, sequence types, K-loci, and identified resistance genes. In addition, a supplementary table listing the accession numbers of all sequences used in the phylogenetic and comparative genomic analyses should be provided. The Materials and Methods section should clearly indicate the origin of these sequences—for example, specifying the databases from which they were retrieved, the criteria for inclusion, and the taxonomic groups represented (e.g., sequences belonging to the Webvirus genus or closely related phages).

7. Biological vs Technical replicates:

Lines 256 and 264 - The notation $n=3$ appears to refer to technical replicates; could the authors please confirm this? In addition, it would be important to clarify how many biological replicates were performed. Biological replicates—ideally derived from independent bacterial cultures prepared on different days and/or conducted in separate 96-well plates—are essential to ensure the reproducibility and robustness of the results.

8. Graph formats:

For transparency in data presentation, and as suggested by Boers 2018 (<https://ard.bmj.com/content/77/6/833.abstract>), "scale of the axes should be chosen carefully. A scale break is mandatory when the range does not include zero. Alternatives include application of log scales". This refers to Figure 2, Figure 3B, Figure 7C and 8C (axes should begin at 0 and a scale break can be included - meaning an axis with 2 segments). In Figure 2B: Please correct the Y-axis to show numbers in percentages instead of scientific notation as it will be easier to understand.

Minor comments (lines are referring to the Maked-up document):

1. Syntax and Minor Language Issues:

While many grammatical issues were corrected, a few unclear phrases remain (e.g., phrases in the abstract and conclusion that still use exaggerated language such as "completely eliminates" infection). A final language check is recommended.

- Line 185 - "period" is misspelled
- Line 216 - correct "removed" to "remove"
- Line 264 - word "served" is missing at the end of the line
- Line 331 - please correct "plaquing" for "plating"
- Line 365 - *Klebsiella* should be italicized
- Line 376 - correct "mixed" to "mixing"

2. Figure 5 Marking:

The phage is now marked in the figure, but the figure legend should explicitly state this for clarity (e.g., "The phage vB_KpnP_Henu1_3 is marked with a red asterisk").

3. Figure 2 Legend:

Needs to be rewritten, including stand-alone information to allow interpretation without reading the Materials and Methods section.

4. Missing information or information that needs to be clarified:

- Line 167 vs Line 190 - which is the correct adsorption time? 5 or 10 min?
- Lines 223-224 - the amendment done on filtered reads above 50bps does not specify the Quality threshold considered. Please include this data.
- Line 244 - Easyfig version used is missing.
- Line 304 - Please add the reference for Graphpad
- Line 315 and 316 - Neither lysis plaque morphology nor phage virion structure are indicative of a virulent lifestyle. Please remove this sentence. This should be approached when the absence of lysogeny genes is discussed.
- Table1 - Please revise capsular types as they are mentioned as K(number) or KL(number) and this nomenclature is not interchangeable. K(number) represents the predicted capsular type, whereas KL(number) corresponds to the K gene locus.

Overall, the revised manuscript has been improved, and the authors demonstrate sincere and effective engagement with reviewer feedback. Adding these new suggestions would make the manuscript clearer, more thorough, and scientifically more robust.

Manuscript ID: Spectrum00931-25R1

Manuscript title: Characterization of the bacteriophage vB_KpnP_Henu1_3 lytic for K1 *Klebsiella pneumoniae* and its therapeutic efficacy in *Galleria mellonella* larvae and mice

Dear editor,

I'm very glad to receive your letter again. Thank you for your thoughtful review of our manuscript and the helpful comments from the reviewers. We are pleased to hear that the manuscript will be potentially acceptable for publication. We agree with the reviewers' suggestions and have revised the manuscript accordingly for your reconsideration. This time, we made every effort to revise the manuscript and were very much looking forward to submitting it for revision. Here below a point-by-point reply to the comments. Hopefully you will find the revised manuscript improved and suitable for publication on *Microbiology Spectrum*.

Yours,
Shanmei Wang

Reviewer #1

-Major methodological concerns:

Q1. Phylogenetic Support for Taxonomy:

While the authors revised the phage's taxonomic assignment, the phylogenetic analysis presented to support this placement is somewhat unclear. Since the manuscript does not specify any information on the sequences included in the phylogenetic trees generated, nor does it mention or highlight which sequences belong to which genus, it is difficult to conclude about the taxonomy. In addition, and very importantly, Siphoviridae is no longer a viral family since the new ICTV classification. A phage can have a siphovirus morphology and be classified within the Caudoviricetes class, but the family to which it

belongs is different and should be correctly mentioned.

We sincerely appreciate your insightful question, which has enhanced our understanding of viral taxonomy. We have now implemented the suggested revisions accordingly. We have listed the accession numbers of the sequence information used for phylogenetic tree construction in the supplementary materials. Regarding the electron microscopy observations of the phage, we have revised the description. "Transmission electron microscopy observations revealed that phage vB_KpnP_Henu1_3 possessed an icosahedral head and siphovirus morphology."

Q2. Statistical analysis:

Despite a new section having been included in the Materials and Methods section, there is no mention of the statistical tests used, neither in this section nor in the figure legends.

This is a must that needs to be included.

Thanks, according to your suggestions, we have added descriptions of the statistical methods used in the figures that include statistical analysis.

Q3. Burst Size Calculation and Consistency:

The reported burst size ($2.94 \times 10^{10} / 1 \times 10^8 = \sim 294$) is inconsistent with the growth curve. Although the authors repeated the experiment, it is still unclear how they determined the number of infected cells at $t=0$, and whether appropriate dilutions and plaque counts were performed to ensure accurate MOI and burst size estimation. The description could benefit from a clearer step-by-step explanation. Even though the initial bacterial culture had a titer of 1×10^8 CFU/ml, the experiment was done at a 0.01 MOI, implying that approximately 1×10^6 CFU/ml were infected at $t=0$. Therefore, considering the burst size calculation (titer of progeny phage = 2.94×10^{10} divided by the titer of infected bacteria = 1×10^6), the burst size should be 2.94×10^4 phages/infected bacteria. This is also in accordance with the one-step growth curve from Fig. 2C, since another formula for calculating the burst size is: PFU/ml at plateau divided by PFU/ml at latent period. Please revise.

Thank you for raising this question. We have consulted a substantial body of literature and synthesized the information. Currently, the methods for calculating the burst size of bacteriophages are primarily conducted through two approaches: one using a high MOI (MOI > 5) infection, and the other using a low MOI infection. The high MOI method ensures that the host bacteria are completely infected in the first round. The low MOI method ensures that most bacteria remain uninfected and can grow normally, thereby providing ample hosts for the replication of the first generation of phages. The burst size is approximately calculated as: $\text{Burst Size} \approx \text{Final phage titer} / \text{Initial number of infected bacteria}$. Whether a high or low MOI is used, it guarantees the complete infection of the initial bacterial population. Since different calculation methods can result in significant variations in the estimated burst size, comparisons of burst sizes between different bacteriophages should be performed under identical experimental conditions. The final phage titer (2.94×10^{10}) represents the total quantity of phages released through multiple rounds of infection. Meanwhile, the entire host bacterial population (1×10^8) was completely infected. Therefore, the calculated burst size is 294. We hope the explanations and answers provided above are satisfactory to you.

Q4. EOP Results Description:

While the authors state that EOP testing was conducted and it has indeed been added to the table, there is no mention of it in Materials and Methods nor the results section.

Thank you for raising this question. We have described both the methodology and results of the Efficiency of Plaquing (EOP) assay in the Materials and Methods and Results sections, respectively. This addition further enhances the completeness of the manuscript.

Q5. Genomic Evidence for Lytic Lifestyle:

The response indicates that lysogeny-related genes were ruled out using online tools, but this analysis is not shown or referenced in the manuscript itself. A brief mention of which tools (e.g., PhageAI, PHACTS, or manual curation of integrase/repressor genes) were used would strengthen this conclusion.

Thanks, according to your suggestions, we have clearly stated the analytical tools used in the Materials and Methods section. Antimicrobial resistance genes and lifestyle traits were predicted using PhageScope.

Wang RH, Yang S, Liu Z, Zhang Y, Wang X, Xu Z, Wang J, Li SC. PhageScope: a well-annotated bacteriophage database with automatic analyses and visualizations. *Nucleic Acids Res.* 2024 Jan 5;52(D1):D756-D761. doi: 10.1093/nar/gkad979.

Q6. Supplementary tables needed:

The detailed information on the bacterial strains, currently included in the revised

Materials and Methods section, would be more appropriately presented as supplementary data. This should include, at minimum: strain designations, dates and sources of isolation, antibiotic susceptibility profiles, and, where available, sequence types, K-loci, and identified resistance genes. In addition, a supplementary table listing the accession numbers of all sequences used in the phylogenetic and comparative genomic analyses should be provided. The Materials and Methods section should clearly indicate the origin of these sequences—for example, specifying the databases from which they were retrieved, the criteria for inclusion, and the taxonomic groups represented (e.g., sequences belonging to the Webervirus genus or closely related phages).

Thank you very much for your feedback. We have now made the revisions as requested. The essential strain information has been included in Supplementary Table S1, details of the bacteriophages used in the comparative genomic analysis are provided in Table 3 in the main text, and the sequence accession numbers used for phylogenetic tree construction are listed in Supplementary Table S2 and S3. The sequences used to construct the phylogenetic tree were all obtained from the NCBI database. The sequences used for phylogenetic tree analysis were all derived from closely related phage sequences.

Q7. Biological vs Technical replicates:

Lines 256 and 264 - The notation $n=3$ appears to refer to technical replicates; could the authors please confirm this? In addition, it would be important to clarify how many biological replicates were performed. Biological replicates—ideally derived from independent bacterial cultures prepared on different days and/or conducted in separate 96-well plates—are essential to ensure the reproducibility and robustness of the results.

Thank you for your question. Here, " $n = 3$ " refers to technical replicates. However, our experiments were also conducted with three independent biological replicates. To ensure accurate representation, we have clarified this description in the Materials and Methods section.

Q8. Graph formats:

For transparency in data presentation, and as suggested by Boers 2018

(<https://ard.bmj.com/content/77/6/833.abstract>), "scale of the axes should be chosen carefully. A scale break is mandatory when the range does not include zero. Alternatives include application of log scales". This refers to Figure 2, Figure 3B, Figure 7C and 8C (axes should begin at 0 and a scale break can be included - meaning an axis with 2 segments). In Figure 2B: Please correct the Y-axis to show numbers in percentages instead of scientific notation as it will be easier to understand.

Thank you very much for your suggestions. We have revised all figures in accordance with the requirements.

-Minor comments:

Q9. Syntax and Minor Language Issues:

While many grammatical issues were corrected, a few unclear phrases remain (e.g., phrases in the abstract and conclusion that still use exaggerated language such as "completely eliminates" infection). A final language check is recommended.

- Line 185 - "period" is misspelled
- Line 216 - correct "removed" to "remove"
- Line 264 - word "served" is missing at the end of the line
- Line 331 - please correct "plaquing" for "plating"
- Line 365 - *Klebsiella* should be italicized
- Line 376 - correct "mixed" to "mixing"

Thank you very much for raising this question. We have corrected the misspelled words in the manuscript.

Q10. Figure 5 Marking:

The phage is now marked in the figure, but the figure legend should explicitly state this

for clarity (e.g., "The phage vB_KpnP_Henu1_3 is marked with a red asterisk").

Thanks, according to your suggestion, we have added the following statement to the figure caption: "The phage vB_KpnP_Henu1_3 is marked with a red asterisk."

Q11. Figure 2 Legend:

Needs to be rewritten, including stand-alone information to allow interpretation without reading the Materials and Methods section.

Thank you very much for your question. We have added a brief description of the experimental methodology to facilitate reader comprehension.

Q12. Missing information or information that needs to be clarified:

- Line 167 vs Line 190 - which is the correct adsorption time? 5 or 10 min?

Thank you very much for your careful review. We have checked the original recorded data and found that this was a writing error. The adsorption time was in fact 10 minutes in all cases, and we have corrected this in the manuscript.

- Lines 223-224 - the amendment done on filtered reads above 50bps does not specify the Quality threshold considered. Please include this data.

Thank you for your valuable comment. The quality threshold applied for retaining reads (including those longer than 50 bp) requires an average Q value of ≥ 20 across 5-bp sliding windows. This criterion ensures that the filtered reads not only satisfy the length requirement but also correspond to a sequencing error rate of $\leq 1\%$. We have added this clarification to enhance the transparency of our data processing workflow.

- Line 244 - Easyfig version used is missing.

Thank you very much for your question. We have now refined the above information.

- Line 304 - Please add the reference for Graphpad

Thank you very much for your question. We have now refined the above information.

- Line 315 and 316 - Neither lysis plaque morphology nor phage virion structure are indicative of a virulent lifestyle. Please remove this sentence. This should be approached when the absence of lysogeny genes is discussed.

Thanks, in accordance with your recommendation, the aforementioned sentence has been deleted.

- Table1 - Please revise capsular types as they are mentioned as K(number) or

KL(number) and this nomenclature is not interchangeable. K(number) represents the predicted capsular type, whereas KL(number) corresponds to the K gene locus.

Thank you very much for your question. Since the specific capsule types of some strains could not be identified based on wzi gene sequencing, they were directly labeled with KL genes in the text. Following your suggestion, we have made revisions, and all strains currently used have clearly defined capsule types.

Re: Spectrum00931-25R2 (**Characterization of the bacteriophage vB_KpnP_Henu1_3 lytic for K1 *Klebsiella pneumoniae* and its therapeutic efficacy in *Galleria mellonella* larvae and mice**)

Dear Dr. Yuan Zhang:

Thank you for the privilege of reviewing your work. Below you will find my comments, instructions from the Spectrum editorial office, and the reviewer comments.

Revision Guidelines

Sincerely,
Leiliang Zhang
Editor
Microbiology Spectrum

Reviewer #1 (Public repository details (Required)):

The accession number of the genome sequence of the phage is provided. However, raw reads have not been deposited in any database, and for clarity, they should.

Reviewer #1 (Comments for the Author):

I appreciate the authors' efforts to revise the manuscript and to address several of the comments raised in the first round of review. The additions to the supplementary tables and clarification of some methods are valuable improvements. However, I remain concerned that several major points have not been adequately resolved, and I believe these need to be addressed before the manuscript can be considered further.

Phylogenetic Classification

Although the description of phage morphology has been corrected, the phylogenetic trees still do not provide sufficient context to support the taxonomic assignment. The trees lack explicit indication of clades or genera, and the manuscript does not clearly state to which family and genus this phage belongs. According to ICTV standards, a full taxonomic classification should be included. Tools such as TaxMyPhage can facilitate this, and I strongly encourage the authors to incorporate such an analysis. Without this, the phylogenetic placement remains incomplete.

Statistical Analyses

The revision mentions the statistical tests used, but important concerns remain. In several experiments with multiple conditions (e.g., stability at different pH values or temperatures), the use of repeated t-tests is not appropriate. In cases where data follow a normal distribution (which must be tested and reported), one-way ANOVA should be applied, followed by suitable post-hoc tests (e.g., Dunn, Bonferroni) to evaluate pairwise differences. The current statistical approach risks inflating type I error. I recommend that the authors seek input from a statistician to ensure correct methodology and transparent reporting.

Burst Size Calculation

The explanation provided still does not satisfactorily address my concerns. The principle of a one-step growth curve is to estimate burst size after a single infection cycle. The current description instead refers to multiple rounds of infection, which is inconsistent with accepted methodology. No reference is provided to justify this approach. As noted previously, the calculation appears to be incorrect. I strongly encourage the authors to consult recent literature for guidance (for example, see: [Nature Microbiology 2025, <https://www.nature.com/articles/s41564-025-02130-4.pdf>]). Without a clear and accurate explanation, the reported burst size is not reliable. In addition, please note that the uploaded figure 2 is a repetition of figure 1, and Figure 2 was not uploaded.

Supplementary Table S1

It is unusual that *K. pneumoniae* strains would show no resistance to any antibiotics. This point remains unclear. Was no antibiogram performed? Or is genomic data unavailable to confirm the presence/absence of resistance genes? This requires clarification, as the current description raises doubts about the strain characterization.

In conclusion, while several revisions have strengthened the manuscript, these unresolved issues-particularly the lack of clear taxonomic assignment, the inappropriate statistical analyses, and the inconsistent burst size calculation-are critical and should be carefully addressed in order to meet the standards of rigor expected for publication.

Manuscript ID: Spectrum00931-25R3

Manuscript title: Characterization of the bacteriophage vB_KpnP_Henu1_3 lytic for K1 *Klebsiella pneumoniae* and its therapeutic efficacy in *Galleria mellonella* larvae and mice

Dear editor,

I'm very glad to receive your letter again. Thank you for your thoughtful review of our manuscript and the helpful comments from the reviewers. We are pleased to hear that the manuscript will be potentially acceptable for publication. We agree with the reviewers' suggestions and have revised the manuscript accordingly for your reconsideration. Here below a point-by-point reply to the comments. Hopefully you will find the revised manuscript improved and suitable for publication on *Microbiology Spectrum*.

Yours,
Shanmei Wang

Reviewer #1 (Public repository details (Required)):

The accession number of the genome sequence of the phage is provided. However, raw reads have not been deposited in any database, and for clarity, they should.

Thank you very much for your question. Providing the raw data is highly beneficial for presenting the experimental results and addressing any subsequent queries from readers. To demonstrate the reproducibility of our sequencing results and the accuracy of our analysis, we have uploaded the raw sequencing data to National Microbiology Data Center. This will enable readers to better reproduce our findings.

Reviewer #1 (Comments for the Author):

I appreciate the authors' efforts to revise the manuscript and to address several of the comments raised in the first round of review. The additions to the supplementary tables and clarification of some methods are valuable improvements. However, I remain concerned that several major points have not been adequately resolved, and I believe these need to be addressed before the manuscript can be considered further.

Phylogenetic Classification

Although the description of phage morphology has been corrected, the phylogenetic trees still do not provide sufficient context to support the taxonomic assignment. The trees lack explicit

indication of clades or genera, and the manuscript does not clearly state to which family and genus this phage belongs. According to ICTV standards, a full taxonomic classification should be included. Tools such as TaxMyPhage can facilitate this, and I strongly encourage the authors to incorporate such an analysis. Without this, the phylogenetic placement remains incomplete.

Thank you very much for your constructive suggestions. With the assistance of the TaxMyPhage tool, we have clarified the taxonomic position of the phage vB_KpnP_Henu1_3, identifying it as a new species within the genus *Webervirus*. The clarification of the taxonomic status of phage vB_KpnP_Henu1_3 provides valuable insights for further exploring its evolutionary relationships with other phages.

Statistical Analyses

The revision mentions the statistical tests used, but important concerns remain. In several experiments with multiple conditions (e.g., stability at different pH values or temperatures), the use of repeated t-tests is not appropriate. In cases where data follow a normal distribution (which must be tested and reported), one-way ANOVA should be applied, followed by suitable post-hoc tests (e.g., Dunn, Bonferroni) to evaluate pairwise differences. The current statistical approach risks inflating type I error. I recommend that the authors seek input from a statistician to ensure correct methodology and transparent reporting.

Thank you for your question. Following your suggestion, we have re-analyzed the data using the one-way analysis of variance followed by Dunnett's multiple comparisons test. This approach ensures the appropriateness of the statistical method used and enhances the accuracy of the experimental conclusions.

Burst Size Calculation

The explanation provided still does not satisfactorily address my concerns. The principle of a one-step growth curve is to estimate burst size after a single infection cycle. The current description instead refers to multiple rounds of infection, which is inconsistent with accepted methodology. No reference is provided to justify this approach. As noted previously, the calculation appears to be incorrect. I strongly encourage the authors to consult recent literature for guidance (for example, see: [Nature Microbiology 2025, <https://www.nature.com/articles/s41564-025-02130-4.pdf>]). Without a clear and accurate explanation, the reported burst size is not reliable. In addition, please note that the uploaded figure 2 is a repetition of figure 1, and Figure 2 was not uploaded.

Based on the references you provided and other relevant literature, our careful review revealed that our previous calculation of burst size might have been affected by secondary infection. This led to an overestimation because the burst size was calculated by dividing the phage titer at the plateau phase by the number of initially infected bacteria. To ensure consistency, we have repeated the one-step growth curve experiments for Phage 1-3 using an improved method informed by the latest references. This included the use of high dilution (1:1000) to prevent secondary infection and progeny phage reinfection, along with optimization of details such as centrifugation temperature and culture medium temperature.

Supplementary Table S1

It is unusual that *K. pneumoniae* strains would show no resistance to any antibiotics. This point remains unclear. Was no antibiogram performed? Or is genomic data unavailable to confirm the presence/absence of resistance genes? This requires clarification, as the current description raises doubts about the strain characterization.

Thank you for your careful review. The absence of antibiotic resistance information for certain strains in Table S1 is due to missing original antimicrobial susceptibility test results. We have now retested the antibiotic susceptibility of these strains and presented the data in the table. The strains we used exhibited varying degrees of resistance to antibiotics, with some being resistant to one antibiotic and others to multiple antibiotics.

In conclusion, while several revisions have strengthened the manuscript, these unresolved issues-particularly the lack of clear taxonomic assignment, the inappropriate statistical analyses, and the inconsistent burst size calculation-are critical and should be carefully addressed in order to meet the standards of rigor expected for publication.

We sincerely thank you for your meticulous review and the journal's stringent requirements, which have not only helped us improve the quality of our manuscript but also deepened our understanding of phage research. In response to the points you raised, we have carefully revised the manuscript and have highlighted all changes using the "Track Changes" feature.

Re: Spectrum00931-25R3 (**Characterization of the bacteriophage vB_KpnP_Henu1_3 lytic for K1 *Klebsiella pneumoniae* and its therapeutic efficacy in *Galleria mellonella* larvae and mice**)

Dear Dr. Yuan Zhang:

Thank you for the privilege of reviewing your work. Below you will find my comments, instructions from the Spectrum editorial office, and the reviewer comments.

Revision Guidelines

Sincerely,
Leiliang Zhang
Editor
Microbiology Spectrum

Reviewer #1 (Public repository details (Required)):

They have already uploaded sequencing raw data, but Bioproject and/or accession numbers are missing (I have checked and the info is indeed uploaded but I suggest adding those numbers for easier access)

Reviewer #1 (Comments for the Author):

The authors have substantially improved the manuscript and have made a commendable effort to address the concerns raised in the previous review round. In particular, I would like to acknowledge the additional work invested in repeating the one-step growth curve experiments and recalculating the burst size using an updated and more rigorous approach. These efforts clearly strengthen the reliability of the results. The manuscript is now close to acceptable; however, a few minor issues should still be addressed:

Provide accession number for raw sequencing reads

The authors indicate that raw reads have been uploaded to the National Microbiology Data Center (NMDC). Please include in the manuscript the corresponding BioProject number or NMDC accession identifier, so that the dataset can be easily located and accessed by readers.

Clarify statistical methodology and normality testing

Since one-way ANOVA has now been used, the manuscript should specify whether the assumption of normality was tested prior to applying this parametric test.

Please state which normality test was applied (e.g., Shapiro-Wilk, Kolmogorov-Smirnov).

If any dataset did not meet normality assumptions, the authors should indicate whether a non-parametric alternative was used, or justify the use of ANOVA despite deviations from normality.

Clarify phylogenetic tree composition or consider removing the figures

Although TaxMyPhage was correctly applied and the taxonomic classification is now clearly reported in the text, the phylogenetic trees remain difficult to interpret.

It is still unclear whether the sequences included represent members of the family, the genus, or a mixture of both.

If the authors decide to keep these figures, they should explicitly indicate the taxonomic scope of the included sequences (e.g., genus-only, genus + outgroups, family-level representatives) and label the clades accordingly.

If this clarification cannot be provided, I recommend removing the phylogenetic trees, as the sequence similarity matrix conveys the evolutionary relationships in a clearer and more concise manner, avoiding redundancy and potential confusion.

Manuscript ID: Spectrum00931-25R4

Manuscript title: Characterization of the bacteriophage vB_KpnP_Henu1_3 lytic for K1
Klebsiella pneumoniae and its therapeutic efficacy in Galleria mellonella larvae and mice

Dear editor,

I'm very glad to receive your letter again. Thank you for your thoughtful review of our manuscript and the helpful comments from the reviewers.

We are pleased to hear that the manuscript will be potentially acceptable for publication.

We agree with the reviewer's suggestions and have revised the manuscript accordingly for your reconsideration.

Here below a point-by-point reply to the comments this time. Hopefully you will find the revised manuscript improved and suitable for publication on *Microbiology Spectrum*.

Yours,

Shanmei Wang

Reviewer #1 (Public repository details (Required)):

They have already uploaded sequencing raw data, but Bioproject and/or accession numbers are missing (I have checked and the info is indeed uploaded but I suggest adding those numbers for easier access)

Thank you very much. We have included the BioProject number (NMDC10020201) for

the raw data upload in the manuscript to facilitate easy access for readers.

Reviewer #1 (Comments for the Author):

The authors have substantially improved the manuscript and have made a commendable effort to address the concerns raised in the previous review round. In particular, I would like to acknowledge the additional work invested in repeating the one-step growth curve experiments and recalculating the burst size using an updated and more rigorous approach. These efforts clearly strengthen the reliability of the results. The manuscript is now close to acceptable; however, a few minor issues should still be addressed:

Thank you very much for your affirmation of our efforts in revising the manuscript and for your contributions to improving the quality of our paper. We have carefully reviewed and addressed the following shortcomings: for instance, we provided the project accession number for the raw data upload, further clarified the statistical analysis methods, and removed the redundant phylogenetic tree analysis.

Provide accession number for raw sequencing reads

The authors indicate that raw reads have been uploaded to the National Microbiology Data Center (NMDC). Please include in the manuscript the corresponding BioProject number or NMDC accession identifier, so that the dataset can be easily located and accessed by readers.

Thank you very much. We have included the BioProject number (NMDC10020201) for the raw data upload in the manuscript to facilitate easy access for readers.

Clarify statistical methodology and normality testing

Since one-way ANOVA has now been used, the manuscript should specify whether the assumption of normality was tested prior to applying this parametric test.

Please state which normality test was applied (e.g., Shapiro-Wilk, Kolmogorov-Smirnov).

If any dataset did not meet normality assumptions, the authors should indicate whether a non-parametric alternative was used, or justify the use of ANOVA despite deviations from normality.

We thank the reviewer for raising this important point. We have now clarified the statistical approach in the revised manuscript. Normality was assessed using the Shapiro-Wilk test. It was found that one/several of our datasets violated the normality assumption ($p < 0.05$). When the sample size is too small ($n=3$), it is also inappropriate to perform a normality test. Consequently, as a non-parametric alternative, the Kruskal-Wallis test was used for these comparisons instead of one-way ANOVA. If a significant difference was detected, Dunn's post-hoc test was employed for pairwise comparisons between the control and each experimental group. We appreciate the opportunity to correct this. This change ensures the appropriateness of our statistical analysis.

Clarify phylogenetic tree composition or consider removing the figures

Although TaxMyPhage was correctly applied and the taxonomic classification is now

clearly reported in the text, the phylogenetic trees remain difficult to interpret.

It is still unclear whether the sequences included represent members of the family, the genus, or a mixture of both.

If the authors decide to keep these figures, they should explicitly indicate the taxonomic scope of the included sequences (e.g., genus-only, genus + outgroups, family-level representatives) and label the clades accordingly.

If this clarification cannot be provided, I recommend removing the phylogenetic trees, as the sequence similarity matrix conveys the evolutionary relationships in a clearer and more concise manner, avoiding redundancy and potential confusion.

Thank you for your valuable feedback. We fully agree with your assessment that the phylogenetic trees could cause confusion due to unclear information, and that the sequence similarity matrix indeed presents the evolutionary relationships more clearly.

Therefore, we have followed your recommendation and removed Figures 5C and 5D (the phylogenetic trees) from the manuscript. Accordingly, we have also eliminated all references to these figures in the text and have strengthened the description of the sequence similarity matrix results. We believe these changes have made the manuscript more logically coherent, eliminated redundancy, and improved its clarity for readers.

Thank you again for helping us enhance the quality of our paper.

Re: Spectrum00931-25R4 (**Characterization of the bacteriophage vB_KpnP_Henu1_3 lytic for K1 *Klebsiella pneumoniae* and its therapeutic efficacy in *Galleria mellonella* larvae and mice**)

Dear Dr. Yuan Zhang:

Your manuscript has been accepted, and I am forwarding it to the ASM production staff for publication. Your paper will first be checked to make sure all elements meet the technical requirements. ASM staff will contact you if anything needs to be revised before copyediting and production can begin. Otherwise, you will be notified when your proofs are ready to be viewed.

Sincerely,
Leiliang Zhang
Editor
Microbiology Spectrum